# A dual-biomarker approach for quantification of changes in relative humidity from sedimentary lipid D/H ratios

Oliver Rach[1,2], Ansgar Kahmen[3], Achim Brauer[4], Dirk Sachse[1]

[1]GFZ – German Research Centre for Geosciences, Section 5.1 Geomorphology, Organic Surface Geochemistry Lab, Telegrafenberg, 14473 Potsdam (Germany)

[2]Institute for Earth- and Environmental Science, University of Potsdam, Karl-Liebknecht-Strasse 24-25, 14476 Potsdam (Germany)

[3]Department of Environmental Sciences-Botany, University of Basel, Schönbeinstrasse 6, CH-4056 Basel (Switzerland)

[4]GFZ – German Research Centre for Geosciences, Section 5.2 Climate Dynamics and Landscape Evolution, Telegrafenberg, 14473 Potsdam (Germany)

*Correspondence to*: Oliver Rach (oliver.rach@gfz-potsdam.de)

**Abstract**

Past climatic change can be reconstructed from sedimentary archives by a number of proxies. However, few methods exist to directly estimate hydrological changes and even fewer result in quantitative data, impeding our understanding of the timing, magnitude and mechanisms of hydrological changes.

Here we present a novel approach based on $\delta^2H$ values of sedimentary lipid biomarkers in combination with plant physiological modeling, to extract quantitative information on past changes in relative humidity. Our initial application to an annually laminated lacustrine sediment sequence from western Europe deposited during the Younger Dryas cold period revealed relative humidity changes of up to 15% over sub-centennial timescales, leading to major ecosystem changes, in agreement with palynological data from the region. We show that by combining organic geochemical methods and mechanistic plant physiological models on well characterized lacustrine archives it is possible to extract quantitative ecohydrological parameters from sedimentary lipid biomarker $\delta^2H$ data.

## 1. Introduction

Predicting future changes in the water cycle using state-of-the art climate models is still associated with large uncertainties (IPCC, 2015). This is because we lack a mechanistic understanding of some of the key processes that influence the water cycle, in particular at regional spatial scales. A better mechanistic understanding of drivers and feedbacks within the hydrological cycle can be achieved from

reconstructing past hydrological changes from sedimentary archives. Stable isotope ratios of meteoric water, expressed as $\delta^{18}O$ and $\delta^{2}H$ ($\delta D$) values are an excellent tool in this respect, because their variability is associated with changes in temperature and source water (Bowen, 2008; Gat, 1996). The isotope ratios of precipitation can be recorded in ice core (Alley, 2000), terrestrial and marine paleoclimate archives through a variety of proxies, such as carbonates (Kanner et al., 2013; von Grafenstein et al., 1999), silicates (Tyler et al., 2008) and lipid biomarkers (Sachse et al., 2012).

Despite their potential, the interpretation of the stable isotope ratios from inorganic and organic proxies often allows only a *qualitative* assessment of past hydrological changes while *quantitative* reconstructions of hydrological changes from isotope proxy data, such as precipitation amount or relative humidity, have been difficult to achieve. This is problematic as quantifiable data are necessary for identifying the mechanistic drivers of past hydroclimate changes as well as their continental scale feedbacks and thresholds for example for vegetation changes. Moreover, quantitative data are needed to test the performance of state-of-the art climate models in simulating past and future changes in the hydrological cycle.

The interpretation of isotope proxies is typically not quantitative because multiple drivers can influence meteoric $\delta^{18}O$ and $\delta^{2}H$ values, hampering the assignment of single quantitative relationships between a hydrologic variable and $\delta^{2}H$ values recorded in a geological archive (Alley and Cuffey, 2001). The increased understanding of the interplay between environmental and plant physiological factors affecting lipid biomarker stable isotope ratios over the last decade (Feakins, 2013; Kahmen et al., 2013a; Kahmen et al., 2013b; Sachse et al., 2009; Smith and Freeman, 2006) has resulted in significant potential for quantitative paleohydrological approaches, exemplified by a reconstruction of seasonality in precipitation and bog surface wetness in a Norwegian peatland (Nichols et al., 2009). Here we take this a step further, combining lipid biomarker hydrogen isotope measurements and plant physiological modeling to constrain the influence of multiple drivers on $\delta^{2}H$ values recorded in organic material and thus allow the extraction of quantitative information about changes in relative humidity from sedimentary archives.

Over the past decade, $\delta^{2}H$ values of lipid biomarkers from photosynthetic organisms have been increasingly used as proxies for reconstructing past changes in the continental hydrological cycle (Feakins, 2013; Rach et al., 2014; Sachse et al., 2012; Schefuss et al., 2011; Seki et al., 2011). In particular *n*-alkanes are ubiquitous in marine and lacustrine sediments and can be preserved over geological timescales (Peters et al., 2007). *n*-Alkanes can be traced back to aquatic or terrestrial sources, where short-chain homologues ($nC_{17}$-$nC_{21}$) are primarily synthesized by algae and aquatic plants (Aichner et al., 2010; Ficken et al., 2000), mid-chain n-alkanes (e.g. $nC_{23}$-$nC_{25}$) by submerged aquatic macrophytes or mosses (Aichner et al., 2010; Ficken et al., 2000; Gao et al., 2011), and long-chain n-alkanes ($>nC_{25}$) predominantly by higher terrestrial plants as a protective leaf wax layer on the leaf surface (Bush and McInerney, 2013; Eglinton and Hamilton, 1967).

Algae and submerged aquatic plants directly use lake (or ocean) water as their hydrogen source for lipid synthesis. $\delta^{2}H$ values from *n*-alkanes from aquatic organisms ($\delta^{2}H_{aq}$) are thus related to the $\delta^{2}H$ value of the water these organisms live in (Aichner et al., 2010; Sachse et al., 2004) offset by a biosynthetic fractionation ($\varepsilon_{bio}$) between water and *n*-alkanes (Sachse et al., 2012) (Eq. (1)). Laboratory

culture studies (Zhang and Sachs, 2007) as well as field studies (Aichner et al., 2010; Sachse et al.,
2004) have resulted in strong linear and nearly 1:1 relationships between source water and $\delta^2H_{aq}$
(Sachse et al., 2012), but have shown that species specific differences in $\epsilon_{bio}$ do exist (Zhang and Sachs,
81   2007).

$$(1) \quad \delta^2H_{aq} = \delta^2H_{precip} + \epsilon_{bio(aq)}$$


Terrestrial plant leaf wax *n*-alkane $\delta^2H$ values ($\delta^2H_{terr}$) have also been found to be linearly correlated to
the organisms source water $\delta^2H$ values, yet not in a 1:1 relationship (Sachse et al., 2012), indicating
additional influences on $\delta^2H_{terr}$ values. Recent greenhouse experiments and field studies have revealed
that in particular the evaporative $^2H$ enrichment of leaf water shapes $\delta^2H_{terr}$ values (Kahmen et al.,
2013a; Kahmen et al., 2013b). Soil water evaporation in the upper soil layers has been shown to be less
significant for $\delta^2H_{terr}$, as plants usually access the deeper, isotopically unenriched, soil layers (Dawson,
1993). As such, $\delta^2H_{terr}$ is affected mainly by the $\delta^2H$ value of plant source water (i.e. precipitation), the
biosynthetic fractionation and leaf water deuterium enrichment ($\Delta^2H_e$) (Eq. (2)).

$$(2) \quad \delta^2H_{terr} = \delta^2H_{precip} + \Delta^2H_e + \epsilon_{bio(terr)}$$


Systematic differences in $\delta^2H_{terr}$ values have been observed for different plant types (especially
between grasses and trees) (Diefendorf et al., 2011; Kahmen et al., 2013b), possibly indicating
differences in either $\epsilon_{bio}$ (Sachse et al., 2012) or the fraction of leaf water used for lipid biosynthesis
(Kahmen et al., 2013b) or yet unidentified factors. As such, vegetation changes in sedimentary records
have been suggested to affect $\delta^2H_{terr}$ values and "vegetation corrections" have been proposed (Feakins,
99   2013).
Since evaporative $^2H$ enrichment of leaf water only affects terrestrial plants but not aquatic organisms,
changes in sedimentary $\delta^2H_{terr}$ (Sachse et al., 2006) can be seen as a record of variations in terrestrial
evaporative $^2H$ enrichment over time. Thus, by combining Eq. (1) and (2) under the assumption that $\epsilon_{bio}$
of both aquatic and terrestrial organisms was constant on the temporal and spatial scales of sedimentary
integration, the difference between $\delta^2H_{aq}$ and $\delta^2H_{terr}$ values should mainly reflect the evaporative $^2H$
enrichment of leaf water (Eq. (3)). Whenever referring to an 'isotopic difference' between two pools
(such as $\Delta^2H_e$) we employ the mathematically correct 'epsilon' formula to calculate differences
between two $\delta$-values (Sessions and Hayes, 2005). For simplicity we use the following expression:

$$(3) \quad \Delta^2H_e = \delta^2H_{terr} - \delta^2H_{aq}$$


Variants of this concept (Sachse et al., 2004) have been used to qualitatively interpret changes in
evapotranspiration through the isotopic difference between $\delta^2H_{terr}$ and $\delta^2H_{aq}$ (i.e. expressed as $\alpha_{TA/wat}$,
$\delta^2H$ $C_{23}$–$C_{31}$ and $\epsilon_{terr-aq}$ (Jacob et al., 2007; Rach et al., 2014; Seki et al., 2011)). With recent progress in
understanding of the determinants of $\delta^2H_{terr}$ values and the existing mechanistic understanding of the
processes governing leaf water evaporative $^2H$ enrichment (Craig, 1965; Kahmen et al., 2011b; Sachse

et al., 2012), we propose a new framework – which we term the dual-biomarker (DUB) approach - to extract quantitative hydrological information, namely changes in relative humidity ($\Delta$rh) from sedimentary records. To illustrate the power of this approach with paleohydrological data, we combine compound-specific hydrogen isotope measurements with plant physiological modeling on a previously published Late Glacial record of $\delta^2H_{aq}$ and $\delta^2H_{terr}$ from sediments of Lake Meerfelder Maar (MFM), Germany (Rach et al., 2014).

## 2. Approach and Model

The key assumptions of the DUB approach are that the difference between terrestrial and aquatic plant derived $n$-alkane $\delta^2H$ values ($\varepsilon_{terr-aq}$) equals evaporative Deuterium enrichment of leaf water (Kahmen et al., 2013b; Rach et al., 2014) over the timescale of sediment integration (i.e. decades in our case) and that $\delta^2H_{lake\ water}$ equals $\delta^2H_{mean\ annual\ precipitation}$, a condition fulfilled for small catchment lakes in temperate environments without any major inflow. Also the temporal delay in transfer of terrestrial $n$-alkanes from source organisms into lake sediment should be below the temporal resolution of the samples, which is fulfilled for sites with a very small catchment area and steep terrain, such as maar lakes. Furthermore we assume that the biosynthetic fractionation ($\varepsilon_{bio}$) is constant for terrestrial and aquatic source organisms on temporal and spatial scales of sedimentary integration (Sachse et al., 2012). We also assume, that palynological data represent lake catchment vegetation so that those can be used to assess source organisms of aquatic and terrestrial $n$-alkanes (Rach et al., 2014; Schwark et al., 2002). To assess the influence of vegetation changes on our reconstructions, we employ two different vegetation corrections based on palynological data, for which we assume that the amount of $n$-alkanes produced by these different plants is equal to the pollen produced by them.

These assumptions and additional data are needed to parameterize the model, therefore we emphasize that a robust application of the DUB model requires a good understanding of the paleolake system and it's environment. As such, the DUB model should only be employed at a site which fulfills the conditions presented above and where a number of additional, well constrained proxy data exist. As of now, this limits the application of the DUB model to precipitation fed, small catchment (ideally maar or crater) lakes in temperate regions.

$\delta^2H_{aq}$ in such systems can be regarded as a direct recorder of growing season average precipitation $\delta^2H$ values and $\delta^2H_{terr}$ values largely reflect leaf water $\delta^2H$ values as has recently been demonstrated for greenhouse and field grown plants (Kahmen et al., 2013a; Kahmen et al., 2013b). Leaf water in turn is a function of the plant's source water and leaf water evaporative $^2H$ enrichment. We argue that soil water evaporation is negligible as recently suggested by several observational studies and a global assessment (Jackson et al., 1996; Jasechko et al., 2013; Kahmen et al., 2013a) and that precipitation is the ultimate water source of aquatic organisms and terrestrial plants. In terrestrial plants however, the source water becomes more enriched in deuterium due to plant transpiration before it is used for lipid biosynthesis. As such, the isotopic difference between $\delta^2H_{terr}$ and $\delta^2H_{aq}$ ($\varepsilon_{terr-aq}$) can be attributed to mean leaf water evaporative $^2H$ enrichment ($\Delta^2H_e$) (Sachse et al., 2004). Based on recent field and greenhouse studies we further assume, that $\varepsilon_{terr-aq}$ captures a growing season signal, probably biased

towards the earlier summer months in temperate climate zones as the majority of leaf waxes is
produced during leaf development with suggested integrational periods between weeks (Kahmen et al.,
2013b; Tipple et al., 2013) and several months (Sachse et al., 2015).

| assumption | explanation |
|---|---|
| $\delta^2H_{lake\ water} = \delta^2H_{mean\ annual\ precipitation}$ | Stable hydrogen isotope composition of lake water equals mean annual stable hydrogen isotope compositions of precipitation (source water), as observed for small catchment lakes in temperate environments (Moschen et al., 2005) |
| $\varepsilon_{terr-aq}$ = leaf water evaporative $^2H$ enrichment | Difference between terrestrial and aquatic plant derived $n$-alkane $\delta^2H$ values equals evaporative Deuterium enrichment of leaf water (Kahmen et al., 2013b; Rach et al., 2014) |
| $\varepsilon_{bio}$ = constant | Biosynthetic fractionation is constant for aquatic as well as terrestrial source organisms on temporal and spatial scales of sedimentary integration (Sachse et al., 2012) |
| no significant delay (i.e. below sample resolution, i.e decades) of terrestrial $n$-alkanes transfer from source organisms into lake sediment | Due to the very small catchment of MFM with steep and wind sheltered crater walls we can assume an almost instantaneous transfer of n-alkanes and pollen from source organisms to lake sediment. Likely autumn leaf litter is the main n-alkane source to the sediment. This is supported by the similar sample to sample (i.e. decadal) variability in the lipid $\delta^2H$ values. If, for example, terrestrial leaf wax n-alkanes would have a substantially longer residence time in the soils before being transported into the lake, then the decadal variability should be much smaller, as the soil would already deliver a more integrated signal into the lake |
| $e_{atm}$ = constant | The atmospheric pressure is inferred from the altitude above sea level (0 meters = 1013 hPa), which remained unchanged. Short term weather related fluctuations (on the order of 100hPa) do not affect the model outcome (see text). |
| $T_{leaf} = T_{air}$ | Leaf temperature equals air temperature on the timescale of sediment integration (decades) (Kahmen et al., 2011b) |
| $\Delta^2H_{wv} = -\varepsilon_+$ | atmospheric water vapor equals equilibrium isotope fractionation between vapor and liquid, as often observed for long-term (several years) time series in temperate climates (Jacob and Sonntag, 1991) |
| no significant influence by Péclet effect | Variations in the Péclet effect are minimal over time in particular for angiosperm species (Kahmen et al., 2009; Song et al., 2013) |
| amount of produced $n$-alkanes from monocots and dicots are almost equal | Both of our vegetation correction approaches assume that palynological reconstructions are representative of leaf wax producing plants and that both monocots and dicots produce similar quantities of $n$-alkanes. |

Table 1: Major model assumptions


The major variables controlling leaf water isotope enrichment are well understood and mechanistic
models have been developed based on the Craig-Gordon evaporation model (Craig, 1965) that allow to
accurately predict or reconstruct leaf water $\Delta^2H_e$ values based on environmental and physiological
input variables (Barbour, 2007; Farquhar et al., 2007; Ferrio et al., 2009; Kahmen et al., 2011b) (Eq.
165   (4))

$$(4)\quad \Delta^2H_e = \varepsilon_+ + \varepsilon_k + (\Delta^2H_{wv} - \varepsilon_k)\frac{e_a}{e_i}$$


$\Delta^2H_e$ is determined by the equilibrium isotope fractionation between liquid and vapor ($\varepsilon_+$), the kinetic
isotope fractionation during water vapor diffusion from the leaf intercellular air space to the
atmosphere ($\varepsilon_k$), the $^2H$ depletion of water vapor relative to source water ($\Delta^2H_{wv}$), and the ratio of
atmospheric vapor pressure and intracellular vapor pressure ($e_a/e_i$) and air temperature ($T_{air}$). In
addition, leaf temperature ($T_{leaf}$), stomatal conductance ($g_s$) and boundary layer resistance ($r_b$) are
essential secondary input variables for the prediction of $e_i$ and $\varepsilon_k$, respectively. Reformulating Eq. (4)
allows expressing $e_a$ as a function of Craig-Cordon variables (Eq. (5)). Since the atmospheric vapor
pressure ($e_a$) can also be calculated based on rh and saturation vapor pressure ($e_{sat}$) (Eq. (6)) we can
merge Eq. (5) and (6) to calculate relative humidity (rh) and to estimate quantitative changes in rh
($\Delta$rh) (Eq. (7)).

$$(5)\quad e_a = \frac{e_i(\Delta^2H_e - \varepsilon_+ - \varepsilon_k)}{\Delta^2H_{wv} - \varepsilon_k}$$


$$(6)\quad rh = \frac{e_a \cdot 100\%}{e_{sat}}$$


$$(7)\quad \Delta rh = \frac{e_i(\Delta^2H_e - \varepsilon_+ - \varepsilon_k) \cdot 100\%}{e_{sat}(\Delta^2H_{wv} - \varepsilon_k)}$$


Equation (7) illustrates that $\Delta$rh can be inferred from a record of past changes in $\Delta^2H_e$ (i.e. a record of
$\varepsilon_{terr-aq}$) if the additional variables $e_{sat}$, $e_i$, $\Delta^2H_{wv}$, $\varepsilon_+$ and $\varepsilon_k$ can be constrained. In the following we discuss
the model parameterizations necessary to apply the DUB approach to estimate quantitative changes in
rh from sedimentary records.

Saturation vapor pressure $e_{sat}$ (Eq. (8)) as well as the equilibrium fractionation factor $\varepsilon_+$ (Eq. (9)) are a
function of temperature (all given numbers and physically variable dependencies within the equations
are transferred from the Péclet-modified Craig-Gordon model by Kahmen et al 2011b and the original
leaf water enrichment model (Craig, 1965; Dongmann et al., 1974; Farquhar and Cernusak, 2005;
Farquhar and Lloyd, 1993)). The atmospheric pressure term ($e_{atm}$), which is also needed for calculation
of $e_{sat}$, describes (mean annual) atmospheric pressure as a function of the elevation above sea level (0
meters = 1013 hPa).

$$(8) \quad e_{sat} = \frac{1.0007 + 3.46 \cdot e_{atm}[hPa]}{1000000} \cdot 6.1121 \cdot exp\left(\frac{17.502 \cdot T_{air}[°C]}{240.97 + T_{air}[°C]}\right)$$


$$(9) \quad \varepsilon_+ = \left[exp\left(\frac{24.844 \cdot 1000}{(273.16 + T_{air}[°C])^2} - \frac{76.248}{273.16 + T_{air}[°C]} + 0.052612\right) - 1\right] \cdot 1000$$


For accurate estimates of $e_{sat}$ as well as $\varepsilon_+$ information on air temperature ($T_{air}$) during the growing
season is thus required. Estimates of past $T_{air}$ variability can be derived from paleotemperature proxy
data to estimate $e_{sat}$ and $\varepsilon_+$ (e.g. chironomids (Heiri et al., 2014; Heiri et al., 2007), MBT/CBT (Blaga et
al., 2013)). In particular chironomid records, thought to represent spring and summer temperatures,
provide an ideal proxy of past mean growing season temperatures in this respect (Heiri et al., 2007).
Note that $e_{sat}$ also depends on the atmospheric pressure (Eq. (8)), which can be estimated from
elevation above sea level and is treated as a constant in the model. Leaf-internal vapor pressure $e_i$ on
the other hand is a function of leaf temperature ($T_{leaf}$). We assume for our calculations that $T_{air}$ is a
good estimate of a growing season average $T_{leaf}$ and $e_i$ can thus be calculated as:

$$(10) \quad e_i = 6.13753 \cdot exp\left(T_{air}[°C] \cdot \frac{18.564 - \frac{T_{air}[°C]}{254.4}}{T_{air}[°C] + 255.57}\right)$$


We are aware that $T_{leaf}$ can exceed air temperature in situations of extreme drought, when transpiration
and evaporative cooling is reduced, or in bright and sunny conditions (Leuzinger and Korner, 2007;
Scherrer et al., 2011). However, on cloudy days as well as on days with wind, $T_{leaf}$ typically equals $T_{air}$
(Jones, 2013). Given the spatial and temporal integration of leaves in sedimentary records (covering
decadal to millennial timescales) it is thus unlikely that single drought events, where $T_{leaf}$ would exceed
$T_{air}$ dominate the overall relationship between $T_{leaf}$ and $T_{air}$ . Recent studies also show that for
temperatures between 15-20°C the $T_{leaf}$ equals $T_{air}$ on seasonal timescales (Kahmen et al., 2011b).
Another parameter affecting leaf water isotope enrichment is the [2]H-depletion of water vapor relative
to source water ($\Delta^2H_{wv}$). In temperate climates liquid water and atmospheric water vapor are often in
isotopic equilibrium, especially when longer (annual to decadal) timescales are investigated (Jacob and
Sonntag, 1991). We therefore assume that $\Delta^2H_{wv}$ equals the equilibrium isotope fractionation between
vapor and liquid $\varepsilon_+$.

$$(11) \quad \Delta^2H_{wv} = -\varepsilon_+$$

In the model, $\Delta^2H_{wv}$ can thus be replaced by $-\varepsilon_+$ (Eq. (11)).
The kinetic isotope fractionation ($\varepsilon_k$) depends on the plant physiological variables stomatal
conductance ($g_s$) and boundary layer resistance ($r_b$) (Eq. (12)) (Kahmen et al., 2011b).

$$(12) \quad \varepsilon_k = \frac{16.4 \cdot \frac{1}{g_s[mol/m^2/s]} + 10.9 \cdot r_b[mol/m^2/s]}{\frac{1}{g_s[mol/m^2/s]} + r_b[mol/m^2/s]}$$


No direct proxies exist to reconstruct these plant physiological variables from sedimentary records, but
paleovegetation data can be used to parameterize the model with biome-averaged values for $g_s$ and $r_b$
that are inferred from modern plants (Klein, 2014). We note that these plant physiological variables
exert only minor control on the model outcome, expected to lie within the analytical error of $\delta^2H$ lipid
measurements (Kahmen et al., 2011b), see also discussion below.
The latest iterations of leaf water models also include a Péclet effect, which describes the ratio of
convectional versus diffusional flow of water in the leaf (Eq. (4))(Kahmen et al., 2011b). However, we
did not include the Péclet effect in our calculations because we assume that variations in the Péclet
effect are minimal over time (Kahmen et al., 2009; Song et al., 2013) in particular for angiosperm
species.
When combining Eq. (9), (10), (11) and (12) with Eq. (7), we obtain a model for $\Delta rh$ (Fig 1) that
requires only four major input variables: $\varepsilon_{terr-aq}$, air temperature ($T_{air}$) as well as literature-derived values
for stomatal ($g_s$) and boundary layer conductance ($r_b$) and one constant parameter ('site altitude above
sea level' for atmospheric pressure ($e_{atm}$)) to calculate $\Delta rh$:

$$(13) \quad \Delta rh = e_i{}'(T_{air}) \cdot \left( \frac{\Delta^2 H_e}{-e_{sat}{}'(e_{atm}, T_{air})\left(\varepsilon_+{}'(T_{air}) + \varepsilon_k{}'(g_s, r_b)\right)} + \frac{1}{e_{sat}{}'(e_{atm}, T_{air})} \right) \cdot 100\%$$


Since we use $\varepsilon_{terr-aq}$ ($=\Delta^2 H_e$) as an input variable, which is representative of leaf water isotope
enrichment above source water and not absolute $\delta^2H$ leaf water values, Eq. (13) predicts changes in rh
($\Delta rh$) but not rh directly. In theory, Eq. (13) would also allow the calculation of rh values directly, if
absolute $\delta^2H_{precip}$ and $\delta^2H_{leafwater}$ was available. The current lack of experimentally determined
biosynthetic fractionation factors for the respective aquatic and terrestrial plants prevents this approach,
but future experimental research may result in robust estimates of $\varepsilon_{bio}$, potentially enabling the
reconstruction of absolute rh values (Zhang et al., 2009).

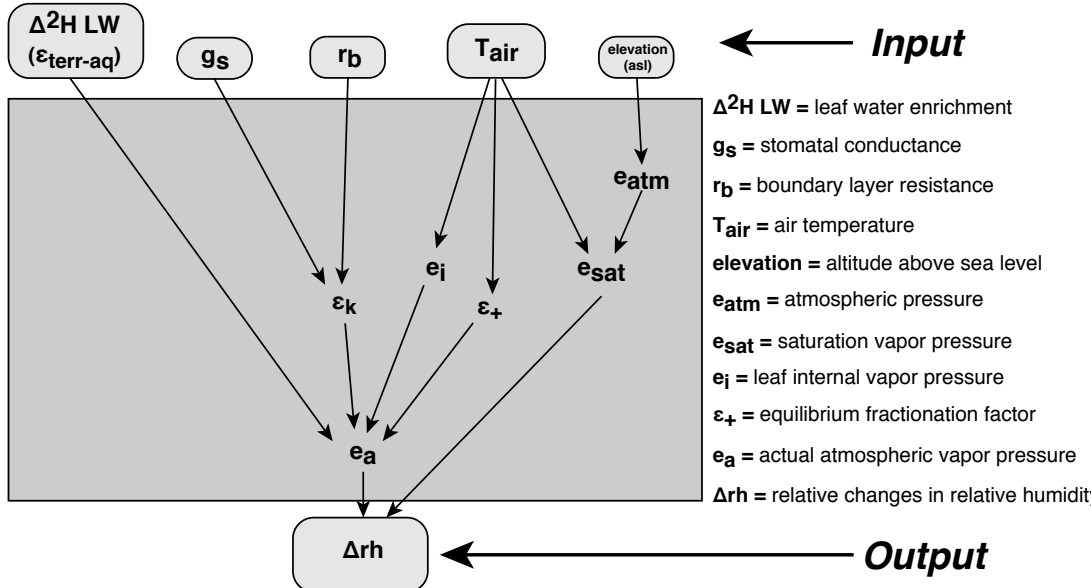

**Fig. 1**: Schematic overview showing the functional relationships between model variables of the DUB approach. Grey boxes on top mark the input parameters while the box size corresponds to the sensitivity of each variable on the result (small box - low influence on Δrh; larger box - higher influence on Δrh)

## 3. Uncertainties and sensitivity tests

### 3.1 Uncertainties

The DUB approach contains different variables (Fig. 1) with specific error ranges which can be quantified. These quantifiable errors (i.e. analytical uncertainties during isotope measurement or paleotemperature determination as well as ranges of values) can be used to set up an error propagation function and finally to provide an error range for the results (e.g. Eq. 16, Appendix). However, additional to these quantifiable uncertainties there are some still some catchment related non-quantifiable uncertainties (see Table 1 and section 2) which can increase the error of the results and therefore need to be taken in consideration before applying to a certain catchment/ record. These unquantifiable uncertainties can however be minimized through the selection of a particular, well characterized lacustrine archive, fulfilling the conditions we outlined under section 2.

### 3.2 Sensitivity tests

To evaluate the robustness of our DUB approach for predicting Δrh in the context of uncertainties, we tested the sensitivity of the model to uncertainties in the four key input variables $T_{air}$, $\varepsilon_{terr-aq}$, $g_s$ and $r_b$. In these sensitivity analyses we used a leaf water model, where all secondary variables ($e_i$, $e_k$, $e_+$, $e_{sat}$) are coupled to the primary input variables $T_{air}$, $T_{leaf}$, $g_s$ and $r_b$ (Kahmen et al., 2011b). We performed this test under a range of dramatically different climatic and ecological settings reflected by the climate conditions of Lista (Norway), Koblenz (Germany), Genoa (Italy) and Perth (Australia) that differ in mean growing season temperatures and prevailing vegetation types. While the vegetation in Norway

and Australia is dominated by conifers and Mediterranean shrubland respectively, the prevailing
vegetation in Germany and Italy are broad leaf tree species. As baseline values for the sensitivity tests
we set $T_{air}$ in the analyses to the growing season mean temperatures of each site, which was 9.4°C,
15°C, 17.2°C and 20.4°C for Lista, Koblenz, Genoa and Perth respectively (IAEA/WMO, 2006). Leaf
water evaporative enrichment $\varepsilon_{terr-aq,}$ ($\Delta^2H_e$) was set to 25‰ (Lista), 35‰ (Koblenz), 45‰ (Genoa) and
55‰ (Perth), which reflects average growing season leaf water enrichment values for the tested
environments (Kahmen et al., 2013a). Base line data for plant physiological variables were biome
typical estimates that we obtained from the literature (Jones, 2013; Klein, 2014): stomatal conductance
($g_s$) for Lista and Koblenz was set to 0.25 mol/m$^2$/s, while for Genoa and Perth the preset values were
0.45 and 0.35 mol/m$^2$/s, respectively (Klein, 2014). Boundary layer resistance ($r_b$) for Lista and Perth
was set to 0.5 m$^2$s/mol, while for Koblenz and Genoa this variable was set to 1.0 m$^2$s/mol (Jones,

290   2013).

The temperature sensitivity tests were performed by increasing and decreasing the respective $T_{air}$
values for a location by 0.5°C, 1°C, 2°C and 5°C (encompassing reconstructed temperature variations
during the last major abrupt climate shift in western Europe – the Younger Dryas period with about 4-
6°C (Goslar et al., 1995; Heiri et al., 2007)). $\varepsilon_{terr-aq}$ ($\Delta^2H_e$ ) values were varied by ± 5‰, 10‰, 15‰ and
20‰ for each location which corresponds to evaporative leaf water enrichment in the test areas (spring
months) (Kahmen et al., 2013a). Plant physiological variables ($g_s$ and $r_b$) were varied by ±0.1, ±0.2,
±0.4 and in maximum by ±0.6 mol/m$^2$/s and ±0.6 m$^2$s/mol, respectively. These tested variations in
plant physiological variables cover the expected variation in $g_s$ and $r_b$ for the local vegetation at the
sites described in the sensitivity analysis.
The sensitivity analyses showed similar results for all four tested environments (Fig. 2). This suggests a
similar behavior of the model under very different climate and ecological conditions. The DUB model
is most sensitive to changes in $\varepsilon_{terr-aq}$ (i.e. $\Delta^2H_e$) and $T_{air}$, while the plant physiological variables ($g_s$, $r_b$)
showed only minor effects on $\Delta rh$ (Fig 2). Specifically, a change of ±20‰ in $\varepsilon_{terr-aq}$ (i.e. $\Delta^2H_e$) resulted
in a change ±20% in $\Delta rh$. A ±5°C change in $T_{air}$ resulted in a 3% change in $\Delta rh$. Varying $g_s$ and $r_b$
within the specified limits caused only changes in $\Delta rh$ of 0.01 to 0.5% (Fig. 2), suggesting low model
sensitivity to plant physiological variables. A sensitivity test with variations in atmospheric pressure
($e_{atm}$) of ±100hPa led to changes in $\Delta rh$ of 0.05%. The difference in calculated $\Delta rh$ for sites with low
(e.g. Lista) and high (e.g. Perth) growing season mean temperature were smaller than the regional
model sensitivity of the different input variables and are therefore negligible. Our sensitivity analyses
shows that the most critical variables for estimating changes in relative humidity with our model are
$\varepsilon_{terr-aq}$ and $T_{air}$ (Fig 2).

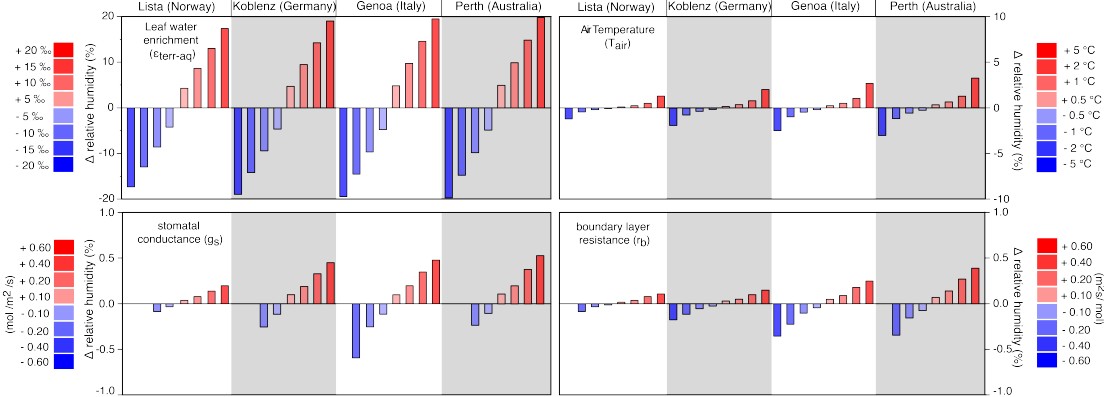

**Fig. 2**: Sensitivity analyses for major model input variables ($\varepsilon_{terr-aq}$, $T_{air}$, $g_s$ and $r_b$) on resulting $\Delta rh$ values tested for four different climatic and ecological environments (Norway, Germany, Italy and Australia). Bars represent the effect on model output ($\Delta rh$) for each tested environment and its variation when the respective input variable will be varied by the marked value. Missing bars (i.e. for negative $g_s$ and $r_b$) results from a bigger (negative) variation than the preset value (below 0).

## 4. Application: Reconstructing quantitative changes in Δrh during the Younger Dryas (YD) in Western Europe

In general, there are two approaches to validate a climate proxy. The most straight forward way is to test the proxy under modern hydroclimate conditions through variations in space or time and compare results with actual instrumental data, either along a modern climatological gradient or over the time period where instrumental data are available. The second possibility is the analysis of a longer time series during a period with otherwise known major changes in the parameter to be tested for.

For testing the DUB model, the first approach is not feasible. While highly resolved (ideally annual laminated) lacustrine sediments from temperate Europe covering the instrumental period (roughly the last 150 years) exist, no major changes in relative humidity occurred during this time. Using only (non-laminated) core top sediments (i.e. only one data point integrating the last decade) would not allow for testing the performance of the DUB approach, which aims to reconstruct relative changes in relative humidity, not absolute data. Testing the DUB approach along a modern climatic gradient is also difficult, because we cannot assume that the source of aquatic biomarkers (in our case $nC_{23}$) is always the same aquatic macrophyte in different lakes and ecosystems (Sachse et al., 2004), i.e. it is unlikely to encounter enough lake systems where the sources of aquatic biomarkers are comparable and which cover a large enough aridity gradient.

Therefore we decided to employ the second approach, i.e. test the proxy during a period of known and significant changes in relative humidity, such as the YD cold period (Rach et al., 2014). The YD the last major abrupt climatic shift in younger earths history (between 12680 years BP and 11600 years BP) characterized by a significant atmospheric temperature decrease of 4-6°C (Goslar et al., 1995; Heiri et al., 2007), a relocation of atmospheric circulation patterns (Brauer et al., 2008) as well as major hydrological changes (i.e. significantly drier conditions) and ecological variations (propagation of grass and reduction of tree vegetation) in western Europe (Brauer et al., 1999a; Litt and Stebich, 1999; Rach et al., 2014). The relocation of atmospheric circulations patterns during Northern Hemispheric cooling

led to drier conditions in western Europe. This forced changes in the regional vegetation composition
(Brauer et al., 1999a; Brauer et al., 2008; Rach et al., 2014). For this period a high resolution record of
changes in $\delta^2H_{aq}$ and $\delta^2H_{terr}$ from a lacustrine archive which fulfills the requirements outlined above
(i.e. precipitation fed, a very small catchment, available palynological and other climate proxy data
(Brauer et al., 1999a; Litt and Stebich, 1999)), Lake Meerfelder Maar (MFM) in western Germany,
exists. The presence of annual varves and a high temporal sampling resolution (decades) allow the
evaluation of the timing of climatic and ecosystem changes - an ideal setting to illustrate the power of
the DUB approach. A detailed description of the record and the available proxy data are given in Rach
et al. (2014). Briefly, the annually laminated sediments of MFM covering the YD period contain
abundant aquatic ($nC_{23}$) and higher terrestrial ($nC_{29}$) lipid biomarkers (n-alkanes) (Fig 3A). Based on
the pollen record, the $nC_{23}$ alkane can be related to the aquatic submerged plant *Potamogeton sp.* and
the $nC_{29}$ alkane to leaves originating from the terrestrial angiosperm trees *Betula* sp. and *Salix* sp. with
input from grasses (Brauer et al., 1999a; Diefendorf et al., 2011). For the DUB approach we use the
isotopic difference between $\delta^2H$ values of the $nC_{29}$ and of $nC_{23}$ alkanes ($\varepsilon_{terr-aq}$) (Fig. 3B) as a measure
for leaf water $^2H$ enrichment ($\Delta^2H_e$).
**4.1 Model parameterization for the MFM application**
**4.1.1 Temperature**
Since no paleotemperature proxy data are directly available for MFM, we use a high-resolution
chironomid based temperature reconstruction from a nearby location, lake Hijkermeer in the
Netherlands (Fig 3C), ca. 300 km N of MFM (see the Appendix). The Hijkermeer record is interpreted
as a record of mean July temperatures for Western Europe with an mean error of about 1.59°C (Heiri et
al., 2007). Since leaf wax synthesis occurs most likely during the early part of the growing season
(spring and summer) (Kahmen et al., 2011a; Sachse et al., 2015; Tipple et al., 2013), the Hijkermeer
record might slightly overestimate spring temperatures. However, when reconstructing Δrh during the
Younger Dryas, it is important that paleotemperature data capture the changes in temperature before
and during that period, rather than absolute temperatures.
**4.1.2 Plant physiological parameters**
We estimated plant physiological variables ($g_s$ and $r_b$) based on literature data from the prevalent
catchment vegetation inferred from available MFM pollen records (Brauer et al., 1999a; Litt and
Stebich, 1999). These suggest that *Betula sp.* and *Salix sp.* were the dominant $nC_{29}$ producing taxa but
that grasses became more abundant during the YD (Brauer et al., 1999a; Litt and Stebich, 1999).
Reported $g_s$ values for these species growing under humid to arid conditions today range from 0.1 to
0.5 mol/m$^2$/s and boundary layer resistance ($r_b$) values from 0.95 to 1.05 mol/m$^2$/s (Klein, 2014;
Schulze, 1982, 1986; Turner, 1984). As input variables for our modified model we therefore used mean
values, i.e. 0.3 mol/m$^2$/s for $g_s$ and 1.0 mol/m$^2$/s for $r_b$. We used the variance of ± 0.2 mol/m$^2$/s for $g_s$
and ± 0.1 mol/m$^2$/s for $r_b$ to calculate the error range of Δrh. We note the low sensitivity of the DUB
model outcome to variability in these variables (see Fig. 2, Appendix), as such that Δrh changes of less
that 0.1% result from varying $g_s$ by 0.4 mol/m$^2$/s or $r_b$ by 0.1 mol/m$^2$/s (Fig. 2).

**4.2 Estimation of uncertainty**

The estimation of uncertainty for Δrh is based on a linear error propagation (Eq. (16) - in the
Appendix) using specific error ranges for the individual input variables. For each input variable we
used their individual reported or estimated error (i.e. for chironomid interfered temperature
reconstruction: ± 1.5°C), for $\varepsilon_{terr-aq}$ the analytical uncertainty (standard deviation) of the respective
biomarker δ$^2$H measurements and for $g_s$ and $r_b$ the observed range of plant physiological parameters
between different species ($g_s$: 0.1-0.5 mol/m$^2$/s, $r_b$: 0.95-1.05 m$^2$s/mol). The resulting average error for
Δrh estimation during the investigated interval is 3.4% (see above and in the Appendix).

**4.3 Model results for the YD period at MFM**

Applying the DUB approach to the Late Glacial MFM record we can for the first time estimate the
magnitude by which rh changed during a distinct period of abrupt climatic change in the past. Our
quantification revealed substantial changes in relative humidity (Δrh) on the order of 30% (Fig 3D)
during the Late Glacial period, some of which occurred on multi-decadal timescales. To better illustrate
these changes we normalized our results to the mean of the period between 12.847 – 12680 BP (mean
Allerød) (Fig 3D), which is thought to have been warmer and moister than the Younger Dryas (Hoek,

406    2009).

In particular, at the onset of the YD at 12.680 years BP, Δrh decreased by 13% +/- 3.4% over 112 years
compared to mean Allerød level (Fig. 3D). During the YD (from 12.680-11.600 years BP) Δrh values
were on average 5% +/- 3.4% lower compared to the mean Allerød level. Furthermore in our high-
resolution dataset we observe a division of the YD into two distinct phases: the first part of the YD
(12.610-12.360 years BP) was characterized by low but relatively constant Δrh (variability between -
8% and -13% and a mean of -10%, compared to Allerød), whereas the variability in Δrh increases after
12360 years BP and ranges between -19% and +2% and a mean of -8% compared to Allerød mean
values (Fig. 3D). Towards the termination of the YD we reconstructed a strong increase in Δrh (up to
+20% above the Allerød level) over only 80 years. This increase started about 100 years before the YD
– Holocene transition at 11.600 BP (Fig. 3D), indicating that hydrological changes lead major
ecosystem changes, which formed the basis for the definition of the YD-Holocene boundary (Brauer et
al., 1999a; Brauer et al., 1999b). The onset of the Holocene was characterized by substantial variability
in Δrh, with a strong increase followed by a decrease to mean Allerød levels 150 years after the
transition. The reconstructed magnitude of changes, i.e. a ca. 9% reduction in rh during the YD
constitutes a shift from an oceanic to a dry summer climate, comparable to the difference in mean
annual rh between Central and Southern Europe today (Center for Sustainability and the Global
Environment (SAGE), 2002; New et al., 1999). The overall temporal pattern of reconstructed Δrh
changes is in good agreement with proxy data from western Europe (Bakke et al., 2009; Brauer et al.,

1999a; Brauer et al., 2008; Goslar et al., 1993), which indicate a shift to drier conditions due to a southward displacement of the westerly wind system chanelling dry, polar air into Western Europe (Brauer et al., 2008; Rach et al., 2014).

Our approach reveals for the first time that substantial changes in rh of up to 20% can take place over very short time scales, i.e. several decades, leading to substantial changes in terrestrial ecosystems. While other proxy data reveal qualitative trends in aridification, our approach can be used to identify hydrological thresholds. Applied to high-resolution records, such as annually laminated lake sediments, the DUB approach can even be used to derive rates of hydrological changes and compare those with associated ecological changes (i.e. pollen records).

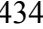

**Fig. 3: (A)** $\delta^2H$ values of aquatic plants ($\delta^2H_{aq}$, blue line) and higher terrestrial plants ($\delta^2H_{terr}$, green line (Rach et al., 2014). **(B)** Terrestrial evapotranspiration ($\varepsilon_{terr-aq}$, orange line) during the Younger Dryas at MFM (Rach et al., 2014). **(C)** Original chironomid based temperature reconstruction from Hijkermeer (NL) (Heiri et al., 2007) (black line with X as data points) and interpolated temperature data for DUB

approach (purple dots). **(D)** Variability of Δrh during the YD cold period at MFM. The data are normalized to mean Allerød level (12.847 – 12.680 years BP). The bold line marks the moving average.

**4.4 The effect of vegetation change on $\varepsilon_{\text{terr-aq}}$ and the estimation of Δrh**

Numerous studies have established that vegetation changes can also affect the sedimentary leaf wax $\delta^2$H record, since significant differences in the net or apparent fractionation ($\varepsilon_{\text{app}}$) between source water and lipid $\delta^2$H values exist among different plant types, in particular between monocot and dicot (all grasses) plants (Kahmen et al., 2013b; Tipple et al., 2013). Since the YD period at MFM was characterized by an increased amount of grasses, we tested, how vegetation changes may affect Δrh reconstructions through the DUB approach. For this we have developed two approaches to "correct" $\delta^2$H$_{\text{terr}}$ values, based on either a constant offset between monocot and dicot $\varepsilon_{\text{app}}$ (Sachse et al., 2012) or a lower sensitivity of grass derived leaf wax $\delta^2$H values to leaf water isotope enrichment (Kahmen et al., 2013b). Both approaches assume that palynological reconstructions are representative of leaf wax producing plants and that both monocots and dicots produce similar quantities of *n*-alkanes.

We used available palynological data to quantify the relative distribution of major tree vegetation (*Betula*, *Salix*) and grasses over the investigated period (Fig. 4B), expressed as the fraction of tress and grasses, $f_{\text{trees}}$ and $f_{\text{grass}}$, assuming that leaf waxes and pollen share a similar transport pathway in this small, constrained crater catchment.

**4.4.1 Correction - case 1 – constant difference in $\varepsilon_{\text{app}}$ between monocots and dicots**

The first vegetation correction for reconstructed leaf water enrichment ($\varepsilon_{\text{terr-aq}}$*) is based on the assumption of a constant offset in biosynthetic isotope fractionation ($\varepsilon_{\text{bio}}$) between trees and grasses. Observational evidence shows that leaf wax lipid $\delta^2$H values ($\delta^2$H$_{\text{terr}}$) from C3 monocots are on average 34‰ more negative that from C3 dicots (non-grasses) when growing at the same site (Sachse et al., 2012). This value is based on an observed mean difference between apparent isotope fractionation (i.e. the isotopic difference between source water and leaf wax *n*-alkanes, $\varepsilon_{\text{app}}$) values of C3 dicots (-111‰) and C3 monocots (-141‰) within a global dataset (Sachse et al., 2012).

The difference between monocot and dicot *n*-alkane $\delta^2$H could potentially affect our modeled Δrh values, especially since an 23% increase in grass abundance in the MFM catchment during the YD has been suggested by pollen studies (Brauer et al., 1999a; Litt and Stebich, 1999).The causes for these differences in $\varepsilon_{\text{app}}$ have been hypothesized to be due to species-specific differences in biosynthetic fractionation (Sachse et al., 2012) or temporal differences in leaf wax synthesis during the growing season (Tipple et al., 2013). Both scenarios would result in a more or less constant isotopic offset between monocots and dicots growing under the same climatic conditions.

Assuming a mean isotopic difference of -34‰ between trees and grasses (Sachse et al., 2012), we calculated a vegetation weighted correction value (-34*$f_{\text{grass}}$) for each data point. This value is then subtracted from $\varepsilon_{\text{terr-aq}}$, and results in the vegetation corrected $\varepsilon_{\text{terr-aq}}$* value (Eq. (14)). Similar

approaches for a pollen based vegetation reconstruction have been recently proposed and applied
(Feakins, 2013; Wang et al., 2013).

$$(14) \quad \varepsilon_{terr-aq}^* = \varepsilon_{terr-aq} - (-34 \cdot f_{grass})$$


**4.4.2 Correction - case 2: different sensitivity to leaf water isotope enrichment in dicot vs.**
**monocot leaf wax $\delta^2$H values**

The second vegetation correction ($\varepsilon_{terr-aq}$**) is based on the assumption that the isotopic difference
between monocot and dicot leaf wax $n$-alkanes is not constant, but dependent on environmental
conditions (Kahmen et al., 2013b). Previous greenhouse studies imply that the difference in $\varepsilon_{app}$
between dicots and monocots is variable depending with a change in humidity conditions (Kahmen et
al., 2013b). In a high humidity climate chamber treatment (80% rh) monocots and dicots showed
similar values for $\varepsilon_{app}$ (-220‰ and -214‰ respectively) whereas in a low humidity treatment $\varepsilon_{app}$ for
monocots was substantially lower compared to dicots (-205‰ and -125‰ respectively) (Kahmen et al.,
2013b), a finding that is in disagreement with the two hypotheses proposed above. Rather, the latter
study hypothesized that grasses use a mixture of enriched leaf water and unenriched xylem water for
lipid synthesis (Kahmen et al., 2013b). This hypothesis would imply that leaf wax $n$-alkane $\delta^2$H values
of monocots do not record the full magnitude of the evaporative leaf water enrichment signal, but only
a fraction (Sachse et al., 2009). A recent greenhouse study on grass derived $n$-alkane $\delta^2$H values of a
broad spectrum of C3 and C4 grasses support this idea (Gamarra et al., 2016). Gamarra et al. suggest
that the differences between $n$-alkane $\delta^2$H values from grasses and $n$-alkane $\delta^2$H values from
dicotyledonous plants are caused by an incomplete transfer of leafwater $\Delta^2$H to the $n$-alkanes. As such,
a sedimentary record of $n$-alkanes derived partly from grasses would also underestimate mean
ecosystem leaf water enrichment. Under dry conditions this fraction was estimated to be ca. 18% for
C3 grasses, based on one grass species (Wheat) studied (Kahmen et al., 2013b). The data from
Gamarra et al. show that for C3 grasses only 38 – 61% of the leaf water evaporative $^2$H-enrichment
signal (depending on the species) was transferred to leaf wax $n$-alkane $\delta^2$H values. To work with a
conservative value and not to overestimate a potential leaf water enrichment signal in grass dervied $n$-
alkane $\delta^2$H values we decided to use the data from Kahmen et al. (2013) for the wheat C3 grass. As
such our correction approach would rather underestimate changes in relative humidity and represents as
such the lower limit of reconstructed changes.
Under the assumption of different sensitivities to leaf water isotope enrichment of $n$-alkane $\delta^2$H values
in monocot and dicot plants (Kahmen et al., 2013b) we developed a correction for $\varepsilon_{terr-aq}$ based on the
experimentally determined mixing ratio between leaf water and unenriched xylem water in wheat, a C3
grass (Kahmen et al., 2013b), essentially by weighing the fraction of grass cover with a factor of 0.18:
(Fig. 4B) (Eq. (15)).

$$(15) \quad \varepsilon_{terr-aq}^{**} = \left(f_{trees} \cdot 1 + f_{grass} \cdot 0.18\right) \cdot \varepsilon_{terr-aq}$$


**4.5 Comparison of results from uncorrected ($\varepsilon_{terr-aq}$) and corrected ($\varepsilon_{terr-aq}$\*, $\varepsilon_{terr-aq}$\*\*) values**

Results from the raw ($\Delta$rh) and both vegetation corrected scenarios ($\Delta$rh\* and $\Delta$rh\*\*) are within the
calculated error range of 3.4% of $\Delta$rh (Fig. 4A) during the Allerød and the Early Holocene, but diverge
by up to 10% during the YD, when C3 grass vegetation was estimated to have increased from 28% to
52% in the catchment of MFM (Fig. 4B). Vegetation corrected results (case 1 Fig. 4A) showed on
average a 7% stronger decrease for $\Delta$rh\* and only a 2% stronger decrease for $\Delta$rh\*\* compared to
uncorrected results. As such $\Delta$rh\*\* values (case 2) are within the error range of uncorrected $\Delta$rh during
the entire record.
Interestingly, both correction approaches, but in particular case 2, place the relatively large variability
in uncorrected $\Delta$rh at the onset and the termination of the YD, where abrupt vegetation changes
occurred. For example, uncorrected $\Delta$rh changes were predicted to be up to 35% during the termination
of the YD, corresponding to the modern gradient between western Europe and the semi-desert areas in
northern Africa (Center for Sustainability and the Global Environment (SAGE), 2002). Vegetation
corrected $\Delta$rh\*\* values were on the order of 20%, seemingly more reasonably representing local Late
Glacial changes (Fig. 4A).
Our analysis shows that vegetation changes have the potential to affect the DUB approach estimates,
but a lack of mechanistic understanding of the causes of the differences in $\delta^2H_{terr}$ between tree and
grass vegetation (Sachse et al., 2012) makes an assessment of the validity of either (or any) correction
approach difficult. Tentatively, the lower variability in $\Delta$rh\*\* within the YD as well as the less
pronounced shift in particular at the onset and termination of the YD (Fig. 4A) provides a more
realistic scenario. But as of now, we regard the differences in predictions as the error of quantitative
predictions from the DUB approach. This uncertainty is larger during periods characterized by
vegetation changes and in our case maximum differences in prediction of $\Delta$rh between the Allerød and
the YD are on the order of 11% (mean Allerød vs mean YD difference between $\Delta$rh and $\Delta$rh\*).

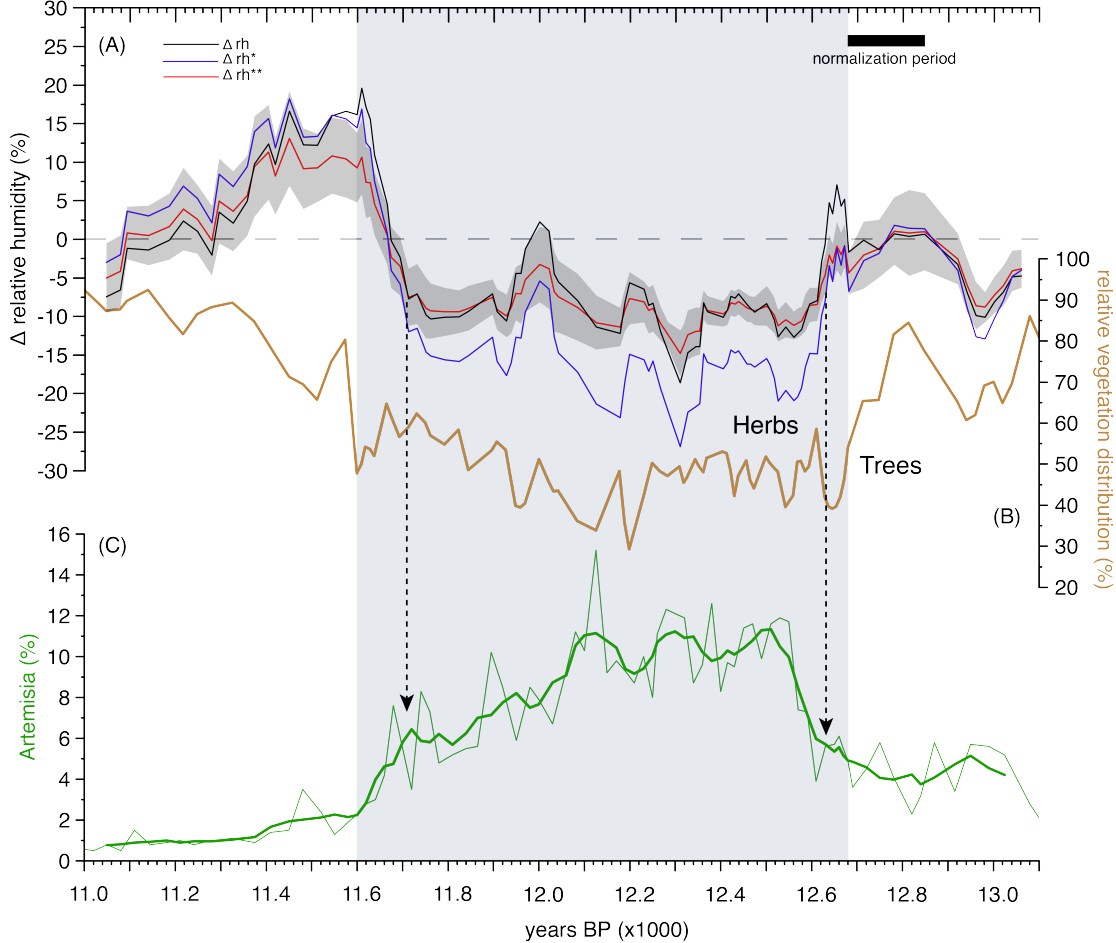

**Fig. 4: (A)** Reconstructed Δrh variability during the YD period (light grey shaded), without vegetation correction (**black line, Δrh**) with vegetation correction assuming a constant offset between C3 dicots and C3 monocots (**blue line, Δrh\*),** with vegetation correction assuming different leaf water sensitivities among grasses and trees (**red line, Δrh\*\***). The shaded area marks the error range for Δrh\*\*. **(B)** relative distribution of trees and grasses in the catchment of MFM during the YD from pollen studies (Brauer et al., 1999a; Litt and Stebich, 1999). **(C)** Occurrence of Artemisia pollen in the catchment of MFM during YD (Brauer et al., 1999a; Litt and Stebich, 1999). Arrows highlight the contemporaneous major changes in Δrh and *Artemisia*.

### 4.6 Comparison of reconstructed Δrh with other proxy data

We can further demonstrate the validity of our approach by direct comparison to other hydroclimate proxies from the MFM record. For example, a classical palynological marker for more arid conditions is *Artemisia* pollen (D'Andrea et al., 2003). In the MFM catchment a prominent increase in the occurrence of *Artemisia* has been used to infer drier conditions during the YD (Fig. 4C) (Brauer et al., 1999a; Bremer and Humphries, 1993; D'Andrea et al., 2003; Litt and Stebich, 1999). When comparing the abundance of *Artemisia* pollen % (note that the *Artemisia* abundance data are not part of the vegetation corrections discussed above) to the DUB Δrh record, we observed striking similarities over the whole of the study period (Fig. 4A,C). Inferred wetter conditions during the second phase of the

YD, or centennial scale excursions to higher Δrh (such as between 12280 and 12170 years BP) go in
line with lower *Artemisia* pollen abundance after 12.100 BP. In fact, both independent datasets show an
inverse, statistically significant relationship (p < 0.001) (Fig. 5A-C), with high *Artemisia* pollen
abundance during periods of low Δrh values (Fig. 4A,C). The correlation between Δrh and *Artemisia* is
higher for vegetation corrected Δrh* and Δrh** (Fig. 5B,C) than uncorrected Δrh and in particular for
Δrh** the variance of the dataset is greatly reduced (Fig. 5C), providing support for the hypothesis that
vegetation changes could have affected the record.

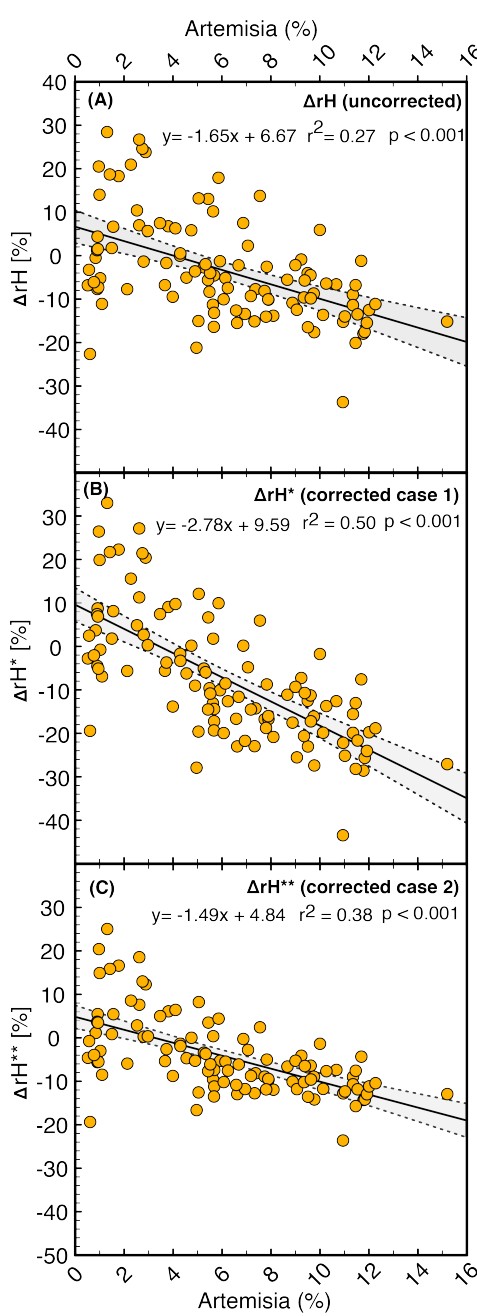


**Fig. 5**: Correlation plots of normalized reconstructed Δrh vs. *Artemisia* population. (**A**) uncorrected Δrh
values vs. *Artemisia*. (**B**) Vegetation corrected Δrh values (Δrh*) vs *Artemisia*. (**C**) Vegetation
corrected Δrh values (Δrh**) vs *Artemisia*.

## 5. Conclusions

We present a novel approach for quantifying paleohydrological changes (i.e. changes in relative humidity) combining sedimentary lipid biomarker $\delta^2H$ values from aquatic and terrestrial lipids with mechanistic leaf water isotope modeling. This dual-biomarker approach (DUB) relies on the observation that aquatic and terrestrial organisms within the catchment of small lakes from temperate climate zones use distinct water sources, namely lake (i.e. precipitation) and $^2H$-enriched leaf water as a source for their organic hydrogen. By taking advantage of the mechanistic understanding of and available models on leaf water isotope enrichment in terrestrial plants, we show it is possible to extract quantitative information about changes in relative humidity from sedimentary records.

Parameterizing and applying the DUB model to a lacustrine lipid biomarker $\delta^2H$ record from western Europe, we find strong and abrupt changes in rh at the onset and the termination of the YD occurring within the lifetime of a human generation. Specifically, our approach showed that shifts in rh of up to 13% +/- 3.4% occurred within only 112 years. This dramatic change corresponds to shifts in average biome rh from oceanic to dry summer climates. Our quantification showed that dry conditions prevailed during the Younger Dryas period with rh being between 8 and 15% lower on average compared to the Allerød, depending on how the possible effect of vegetation changes is accounted for. The pattern but also the magnitude of our rh reconstruction agrees well with other proxy data, such as the increase in the abundance of specific taxa adapted to dry conditions (e.g. *Artemisia*) during that time period.

Our analyses shows that the DUB approach is capable of quantifying past hydrological changes in temperate environments, when additional proxy data, especially on vegetation distribution and paleotemperature exist. We suggest that this approach can be particularly valuable in the future for the validation of climate models and to better understand uncertainties in predictions of future hydrological change under global warming. However, we stress that the DUB approach relies on a number of assumptions and is currently limited by our incomplete understanding of processes affecting the transport and deposition of in particular terrestrial biomarkers from their source to the sedimentary sink. To minimize the arising uncertainties, this approach should only be applied to small catchment lake systems which are fed by precipitation in temperate climate zones, when biomarker sources can be constrained by paleovegetation data (such as palynological records). It is particularly crucial to constrain the aquatic biomarker source, but in principle any aquatic lipid biomarker (macrophyte, algal) could be employed. Our reconstruction provides reasonable values of rh changes during the YD cold period, which are in agreement with ecosystem changes in the region. As such, the present approach provides a first step towards quantitative paleohydrological reconstructions.

**Appendix**

**Error propagation**

The uncertainty estimation (Δf, Eq. (16)) for the reconstructed Δrh variability is based on a linear error propagation, which is the most conservative method for error estimations. This Method does not require the same kind of the considered errors and provides therefore the possibility to combine different kinds of errors with their specific ranges (i.e. measuring error, counting error, etc.). The individual error ranges of the independent variables in our approach arise from different sources such as analytical errors (chironomid interfered temperature reconstruction: ± 1.5°C), observed variations of plant physiological parameters between different species (stomatal conductance: 0.1-0.5 mol/m$^2$/s, boundary layer resistance: 0.95-1.05 m$^2$s/mol) and standard deviation of $\delta^2$H measurements of terrestrial and aquatic *n*-alkanes.

The specific uncertainty for $\varepsilon_{terr-aq}$** was preliminary determined by a separate error propagation using the (analytical) standard deviation of the triplicate measurements of the sedimentary *n*-alkane $\delta^2$H values as well as the plant derived *n*-alkane $\delta^2$H measurements by Kahmen et al 2013. The results of these separate error estimation were integrated into the general error estimation of Δrh**.

In contrast to the linear error propagation a less conservative method (Gaussian error propagation) requires a similarity of the errors, i.e. all errors are measurement or counting errors, which is not the case in this study. The mean error when using the Gaussian method is however only 3.2% and therefore only 0.2% smaller than the calculated error using the linear propagation.

$$(16) \quad \Delta f = \left| \frac{\partial rh}{\partial \varepsilon_{terr-aq}} \right| \cdot \Delta \varepsilon_{terr-aq}^{**} + \left| \frac{\partial rh}{\partial r_b} \right| \cdot \Delta r_b + \left| \frac{\partial rh}{\partial g_s} \right| \cdot \Delta g_s + \left| \frac{\partial rh}{\partial T_{air}} \right| \cdot \Delta T_{air}$$

**Temperature data**

The temperature data used for the DUB model parameterization of the MFM case were taken from ref. 35 and constitute reconstructed summer temperatures based on chironomid analyses from Hijkermeer (NL) (Heiri et al. (2007)), which, to our knowledge, constitutes the closest lateglacial paleotemperature record to the MFM site (distance 311km). However, the dataset of the Hijkermeer consists only of 37 data-points between 13.000 BP and 11.000 BP with a temporal resolution varying between 26 to 167 years /sample. Therefore, we determined a new equidistant time-series for the temperature data, fitting data-volume and temporal resolution of our $\Delta^2$H$_e$ record from MFM (106 data-points with an 8 to 33 year-resolution). For calculating the equidistant time series we were using method "interpl" with the specification "linear" in MATLAB (version R2010b).

**Vegetation data**

Information about Lateglacial vegetation-cover in the catchment area of MFM is based on palynological analyses (Brauer et al. (1999), Litt & Stebich (1999)). We used Pollen percent data also for determining the vegetation distribution between trees and grasses for each datapoint. For using these vegetation data in our model it was necessary to determine an equidistant time-series according to age model of our $\Delta^2H_e$ values. For calculating these time series we used also method "interpl" with the specification "linear" in MATLAB (version R2010b).

**Author contributions**

Oliver Rach conducted model modifications, calculations and wrote the paper. Ansgar Kahmen provided the basic leaf water enrichment model and was responsible for plant physiological part and contributed in writing the paper. Achim Brauer was responsible for lake coring, provided the chronology and stratigraphy for Younger Dryas hydrological reconstruction and wrote the paper. Dirk Sachse conceived the research, acquired financial support and wrote the paper.

**Competing financial interests**

The authors declare no competing financial interests.

**Acknowledgements**

This work was supported by a DFG Emmy-Noether grant (SA1889/1-1) and an ERC Consolidator Grant (No. 647035 *STEEP*clim ) to D.S. It is a contribution to the INTIMATE project, which was financially supported as EU COST Action ES0907 and to the Helmholtz Association (HGF) Climate Initiative REKLIM Topic 8, Rapid climate change derived from proxy data, and has used infrastructure of the HGF TERENO program.

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
