# Peer review of "A dual-biomarker approach for quantification of changes in relative humidity from sedimentary lipid D/H ratios"

_Climate of the Past, 2017_

## Referee Comment (RC1) · Anonymous Referee #1 · 16 Feb 2017

Rach et al. present a mechanistic leaf-water isotope model to quantify for the first time past changes in relative humidity using lake sedimentary leaf-wax dD records. The model, which is suited for reconstructions from temperate regions where lake water evaporation is negligible, is tested using previously published proxy data from the type site of Meerfelder Maar (MFM) in Western Germany, with a focus on the period spanning the climate transitions into and out of the Younger Dryas stadial.

Personally, I'm glad to see this work finally coming along after several years of development. I think this study constitutes a long overdue leap towards a more quantitative estimate of lipid-based stable water isotope reconstructions, which will ultimately be of interest to a broad readership including organic geochemists, paleobotanists, paleoclimatologists and climate modellers.

The construction of the model leans upon a large number of unavoidable assumptions involving both plant physiology and environmental conditions. While the majority of the physiological assumptions are reasonably satisfied through empirical evidence, I think some of the assumptions surrounding past environmental and climate conditions are less convincing. Nonetheless, due to the lack of sound proxies that allows reconstructing parameters such as local vegetation cover, atmospheric pressure, wind strength and seasonal air temperature (among others), the model formulation presented in this paper should be considered –overall– as good as it can get.

However, I think there is still much room for improvement and I have a number of serious concerns, primarily involving the methodological approach, the data-uncertainty treatment, and the contents of the paper that the authors should address before publication.

Main comments

1) I am very surprised the authors decided to design this study directly around down-core data without attempting validation of their model using core-top samples from MFM. I think the model should first be tested using lipid $\delta$D measurements from surface sediments in tandem with meteorological observations before a deglacial Rh reconstruction is discussed. Whether this unusual direction was taken intentionally or not, I think the authors should provide some explanations.

2) One of the main assumptions upon which the results depend is that pollen records are indicative of the local vegetation cover and that pollen counts scale linearly with waxes concentration in MFM sediments. If this were the case, one would expect to observe a clear correlation between the abundance of n-alkanes and pollen records. I would therefore like to see the distribution of n-alkanes plotted together with selected pollen data (e.g. relative distribution of trees, shrubs, herbs). Ideally, the authors should also present some correlation statistics or at the very least a visual correlation using

for instance the average n-alkane chain length (and other chain length ratios) to make a qualitative distinction between graminoids and woody plants that would confirm the authors' hypothesis.

3) The authors argue that monocots integrate a varying evaporative $\delta$D enrichment signal in response to changes in humidity conditions. Following this reasoning, the mixing ratio between leaf water and un-enriched xylem water in C3 grasses will change as a function of the aridity level in the catchment. Therefore, the weighing of the fraction of grass cover used to correct $\varepsilon$terr-aq should also vary over time, whereby a much higher fraction of C3 grass integrated the leaf-water evaporative $\delta$D enrichment during the Late Allerød (AL) and Early Holocene, when climatic conditions were presumably wetter. The authors instead apply a constant weighing factor of 18% throughout the record in their **correction. This issue should be addressed and the weighing factor should vary over time according to the humidity conditions as inferred using an independent proxy (for instance Artemisia pollen percentages).

4) Despite the authors' effort to address all the potential errors that accompany their data, there is a large source of uncertainty that has been neglected and that has been allegedly circumvented by simply including the analytical uncertainty of the $\delta$D measurements. This source of uncertainty is the variability associated with the apparent isotope fractionation values of C3 dicots and C3 monocots as observed for modern plants (e.g. Sachse et al., 2012). This variability should generally be accounted for when estimating Rh. It should also be taken into consideration when calculating the vegetation corrected $\varepsilon$terr-aq*, as the authors simply apply the mean $\varepsilon$app difference between dicots and monocots disregarding the related uncertainty, which can be quite substantial (as high as 30‰ at 1 sigma) for each plant group. While I understand that quantifying this uncertainty would disproportionally increase the error bounds of their Rh reconstruction, I think the authors should at least discuss this issue more openly in the text.

Furthermore, along these lines I believe that the authors should attempt at estimating how well the pollen records represents the local vegetation cover and include this source of uncertainty in the error propagation equation.

5) I am quite sceptical about the use of chironomid-based temperature reconstructions to infer air temperatures. Although I recognize that as yet lacustrine midge assemblages are one of the best proxies for summer air temperature, they have shown to incorporate a number of environmental signals apart from air temperature alone, most importantly catchment vegetation, nutrient levels, lake depth and seasonality (Eggermont and Heiri, 2012; Luoto, 2010). It is therefore likely that this biological proxy was very sensitive to the major environmental and seasonality shifts that occurred at the onset and termination of the YD. Especially, colder and longer winters during the YD might have resulted in relatively colder surface water temperatures during the growing season relative to the air (e.g. replenishment of the local aquifer via snow thawing), in contrast to the preceding warm AL phase. As chironomids are sensitive to water temperatures, it is hard to say to what extent the proxy is biased towards colder temperatures during the YD, not mentioning that the authors use a Dutch record thus making impossible to assess local versus regional factors. I think some of these issues should be openly discussed in the paper.

Moreover, if I remember correctly the chronology that underpins the record from Hijkermeer is based on 14C dating of regional pollen boundaries that have been dated elsewhere. This implies that the temperature record comes with fairly large age uncertainties as compared to the proxy records at MFM, where the chronology is much more accurate. Assuming that the alignment between Hijkermeer pollen records and the regional pollen stratigraphy is precise, I wonder if the author can include in their error propagation estimate the age uncertainty associated with the temperature reconstruction. Even though I understand temperature plays only a minor role in the DUB model, in my opinion, this would result in a much more rigorous estimation of the true uncertainties that accompany the reconstructed Rh.

In addition, I suggest that for reference the authors plot the temperature record together

with the Rh reconstruction.

6) Since the authors decided to reconstruct Rh across the Younger Dryas, I think it would be appropriate to briefly present the status of the knowledge on this climate event (as well as to better frame their results into a paleoclimatological context). I suggest to mention the ocean–sea-ice–atmosphere mechanisms that would explain the climate variability observed in European climate reconstructions during this period (e.g. Brauer et al., 2008; Lane et al., 2013; Muschitiello et al., 2016; Rach et al., 2014). I would also recommend that the authors discuss the current understanding of hydro-climatological variability at the onset and termination of the YD in Europe based on the lake sedimentary $\delta$D and $\Delta\delta$Dterr-aq reconstructions available so far.

Specific comments

The DUB is based on a number of important assumptions that are discussed along the text (I counted at least 14 in the paper, some of which are "hidden" between the lines). I wonder if the author can provide a summary of these assumptions in the form of a table to facilitate rapid screening.

Similarly, I suggest that all the model parameters, fixed variables, and sensitivity tests are summarised in separate tables. As they are, these information are hard to piece together.

L170: The authors adopt a constant atmospheric pressure value in their model. I would first like to know what value they use and why? Secondly, I would like to know how sensitive their model is to this parameter. Climate modelling studies have shown that there were considerable summer sea level pressure changes over Northern-Central Europe from the late AL to the YD (Menviel et al., 2011; Muschitiello et al., 2015). I would therefore be inclined to apply different sea-level pressure values across the deglaciation. Perhaps the authors can comment on this and openly discuss these problems in the paper.

L176-182: Is there any empirical value that allows calculating Tleaf as a function of Tair? For the reasons I outlined in the previous comments, assuming that it is constantly (and equally) cloudy and/or windy at MFM during both AL and YD does not necessarily hold.

L487-488: A number of studies have shown a bi-partite structure of the YD with relatively drier conditions in Northern, Central and Southern Europe during the Early YD and relatively drier conditions during the Late YD (Bakke et al., 2009; Bartolomé et al., 2015; Lane et al., 2013). It surprises me that this mid-YD transition is not clearly captured in the Rh reconstruction at MFM. Although the authors claim that the record reveals "centennial scale excursions to higher $\Delta$Rh after 12.100 BP" I struggle to see any appreciable change in Rh variability. Critically, a marked shift in Rh after the mid-YD transition would support the reconstructed Rh, since virtually no significant vegetation shift had occurred during the YD and therefore the modelled Rh is independent of potential influences from local vegetation changes during this period. However, I must acknowledge that so far the mid-YD transition has been inferred only using qualitative or indirect hydro-climate proxies and thus a net shift from dry to wet conditions in Europe still requires conclusive evidence. Perhaps these issues can be briefly addressed in an apposite YD section of the paper (please see main comment on YD background discussion).

L491-494: How does the percentage of shrub pollen vary with respect to the percentage of tree and herb. If there is a strong covariance between shrubs, trees and herbs then it is not surprising that both the vegetation-corrected (using grass and tree+grass, respectively) Rh reconstructions correlate with the Artemisia pollen percentage (i.e. included in the shrubs pollen record) better than the uncorrected Rh. I would therefore like to know the level of covariance between the distributions of trees, grasses and shrubs at MFM. In addition, I would recommend that the authors include the relative shrub pollen percentages in Figure 3 for reference (please note that in the same figure either tree or herb distributions have not been plotted).

I also wonder whether it is possible that the improved fit between the corrected Rh and Artemisia data (Figure 4) merely stems from subduing the Rh series variability when applying the vegetation correction.

I believe that the paper would benefit from including some selected pollen diagrams (as supplementary material for example). Analogously, I think the original $\delta$D and $\Delta\delta$Dterr-aq records should also be presented in Figure 2 or 3.

I also recommend that the author consider to include as Figure 1 their conceptual overview model of the hydrogen-isotopic relationship between source water and sedimentary lipids (Figure 6 in Sachse et al. (2012) and Figure S6 in Rach et al. (2014)) to illustrate the initial formulation steps of the DUB model.

Line-by-line comments

L77-78 and 87-88: In equations (1) and (2) please specify that the terms $\varepsilon$bio refer to terrestrial and aquatic components, respectively (i.e. $\varepsilon$bio (terr) versus $\varepsilon$bio (aq)).

L159: The term esat "Saturation vapour pressure" should be introduced at line 148.

L172: Missing "of" after "function".

L243-244: Please provide reference for this statement.

L392: The line in brackets should start with small letters.

L416: "...low humidity treatment": how much?

L462: "...provides are more..." should read "...provides a more..."

Figure 1: The data plotted in the upper-right panel are not in scale with the data presented in the upper-left panel. Please adjust.

Figure 3: Either the tree or shrub relative distribution is missing from the figure.

References

Bakke, J., Lie, Ø., Heegaard, E., Dokken, T., Haug, G.H., Birks, H.H., Dulski, P., Nilsen, T., 2009. Rapid oceanic and atmospheric changes during the Younger Dryas cold period. Nature Geoscience 2, 202–205. doi:10.1038/ngeo439

Bartolomé, M., Moreno, A., Sancho, C., Stoll, H.M., Cacho, I., Spötl, C., Belmonte, Á., Edwards, R.L., Cheng, H., Hellstrom, J.C., 2015. Hydrological change in Southern Europe responding to increasing North Atlantic overturning during Greenland Stadial 1. Proceedings of the National Academy of Sciences 201503990. doi:10.1073/pnas.1503990112

Brauer, A., Haug, G.H., Dulski, P., Sigman, D.M., Negendank, J.F.W., 2008. An abrupt wind shift in western Europe at the onset of the Younger Dryas cold period. Nature Geoscience 1, 520–523. doi:10.1038/ngeo263

Eggermont, H., Heiri, O., 2012. The chironomid-temperature relationship: Expression in nature and palaeoenvironmental implications. Biological Reviews 87, 430–456. doi:10.1111/j.1469-185X.2011.00206.x

Lane, C.S., Brauer, A., Blockley, S.P.E., Dulski, P., 2013. Volcanic ash reveals time-transgressive abrupt climate change during the Younger Dryas. Geology 41, 1251–1254. doi:10.1130/G34867.1

Luoto, T., 2010. Spatial and temporal variability in midge (Nematocera) assemblages in shallow Finnish lakes ($60-70°$ N): community-based modelling of past environmental change. PhD Thesis (Helsingin Yliopisto).

Menviel, L., Timmermann, A., Timm, O.E., Mouchet, A., 2011. Deconstructing the Last Glacial termination: The role of millennial and orbital-scale forcings. Quaternary Science Reviews 30, 1155–1172. doi:10.1016/j.quascirev.2011.02.005

Muschitiello, F., Lea, J.M., Greenwood, S.L., Nick, F.M., Brunnberg, L., MacLeod, A., Wohlfarth, B., 2016. Timing of the first drainage of the Baltic Ice Lake synchronous with the onset of Greenland Stadial 1. Boreas 45, 322–334. doi:10.1111/bor.12155.

Muschitiello, F., Pausata, F.S.R., Watson, J.E., Smittenberg, R.H., Salih, A.A.M., Brooks, S.J., Whitehouse, N.J., Karlatou-Charalampopoulou, A., Wohlfarth, B., 2015. Fennoscandian freshwater control on Greenland hydroclimate shifts at the onset of the Younger Dryas. Nature Communications 6, 1–8. doi:10.1038/ncomms9939

Rach, O., Brauer, a., Wilkes, H., Sachse, D., 2014. Delayed hydrological response to Greenland cooling at the onset of the Younger Dryas in western Europe. Nature Geoscience 7, 109–112. doi:10.1038/ngeo2053

Sachse, D., Billault, I., Bowen, G.J., Chikaraishi, Y., Dawson, T.E., Feakins, S.J., Freeman, K.H., Magill, C.R., McInerney, F. a., van der Meer, M.T.J., Polissar, P., Robins, R.J., Sachs, J.P., Schmidt, H.-L.,

Sessions, A.L., White, J.W.C., West, J.B., Kahmen, A., 2012. Molecular Paleohydrology: Interpreting the Hydrogen-Isotopic Composition of Lipid Biomarkers from Photosynthesizing Organisms. Annual Review of Earth and Planetary Sciences 40, 221–249. doi:10.1146/annurev-earth-042711-105535

---

## Referee Comment (RC2) · S. Feakins (Referee) · 17 Feb 2017

General Comments

The authors present a paleoclimate record from laminated sediments. They extract a dual hydrogen isotope record from two homologues of n-alkanes, each thought to be derived from aquatic algal production and terrestrial plant production, respectively. The attempt is to move beyond qualitative interpretations to develop quantitative interpretations of relative humidity. The approach is reasoned, the climate result important and the manuscript should be suitable for publication in CP after appropriate revisions. The manuscript is generally well written, although reviewer 1 has raised extensive comments about climatic interpretations and Ebio interpretations including the question of

why no modern calibration was attempted as part of proof-of-concept. I will not repeat any of these comments but will confine my review to raising a technical but substantive issue that undermines the quantitative claims at present, by introducing non-trivial arithmetic errors. If the authors revise their approach with the correct arithmetic formulations, the approach will be a quantitatively robust contribution. In my opinion this fundamental revision of the calculations is required before further consideration for publication. Without such correction, the introduction of non-trivial arithmetic errors represent an impediment to accurate climatic interpretations using a widely-used paleohydrological proxy.

Specific Comments

Line 74 "1:1" represents a misunderstanding of the mathematical implications of the relative isotope terms. A 1:1 line would not be expected for a fixed fractionation. Please review fractionation terms as indicated in Section 3.1 of Sessions and Hayes (2005). For the slope y = mx + c, y = alpha*x + epsilon, where epsilon = alpha - 1. The difference term approximation is acceptable when alpha is between 0.95 to 1.05 as is often the case for carbon or oxygen but is inappropriate for hydrogen, when values of alpha may be 0.8 to 0.9 for many plants.

Similarly, equations 1-4 are not in the correct form, they are combined as though they were difference terms, when this is not appropriate for the relative calculations implicit in epsilon terms. Even if the results are trivially different (which they appear not be), this approximation is not advisable because it builds misunderstanding that is likely to propagate through the literature.

Equations of a similar form have been published for oxygen isotope considerations. The error introduced is trivial for the smaller fractionations associated with the smaller relative mass difference between 16O and 18O (Kahmen et al., 2011), but it matters when that approach is extended to 1H and 2H where the relative mass difference is 8-times higher and the fractionations commensurately larger. Admittedly the algebra will

be considerable, but the formulation could be provided in a spreadsheet rather than as equations within the text. The authors must at minimum account for the uncertainties introduced by the mathematical approximations, but ideally they will revise their equations accordingly, given the magnitude of errors introduced are non-trivial for their RH interpretations.

Technical Corrections

I have performed some simple calculations with example input data to illustrate the magnitude of the arithmetic errors introduced by the incorrect formulation based on difference terms in Eqns. 1-4. I also illustrate that the 1:1 line is not the expected result of a fixed fractionation. The output is provided here (Figure 1) and the Excel file supplied as Appendix.

References

Kahmen, A., Sachse, D., Arndt, S.K., Tu, K.P., Farrington, H., Vitousek, P.M. and Dawson, T.E. (2011) Cellulose delta O-18 is an index of leaf-to-air vapor pressure difference (VPD) in tropical plants. Proc. Natl. Acad. Sci. U. S. A. 108, 1981-1986.

Sessions, A.L. and Hayes, J.M. (2005) Calculation of hydrogen isotopic fractionations in biogeochemical systems. Geochimica Et Cosmochimica Acta 69, 593-597.

Please also note the supplement to this comment:
http://www.clim-past-discuss.net/cp-2017-7/cp-2017-7-RC2-supplement.zip

[Figure]

**Fig. 1.** Reviewer demonstrations of introduced errors by the difference approach

---

## Author Comment (AC1) · 21 Feb 2017

Dear Sarah, we want to thank you for the input on our paper. We understand your concern, and we cannot agree more, in particular for quantitative reconstructions we need to make sure no arithmetic errors are introduced. However, we want to clarify that we did all calculations using the correct mathematical approach (i.e. using the 'epsilon' formula for calculations with delta values) as emphasized by (Sessions & Hayes 2005). Therefore, our calculations would arrive at the exact same value as you do in your example calculation (i.e. "reviewer calculation"), and not at values you suggest if we would have used the incorrect mathematical expression (i.e. "author calculation" in the example spreadsheet added to the comment).

As such, no arithmetic errors were introduced into our model calculations.

For the estimation of $\Delta$rH the only relevant equation is eq. 3, where we use the isotopic difference between terrestrial and aquatic biomarkers ($\delta$2Hterr - $\delta$2Haq) as a measure of mean leaf water enrichment above source water ($\Delta$2He), since this parameter goes into our final model (eq. 13). $\Delta$2He was calculated using the correct (epsilon) formula. These values are also equal to the $\varepsilon$terr-aq data from (Rach et al. 2014). We adapted the use of '$\Delta$' instead of '$\varepsilon$' here, based on the common use in the plant physiological literature (where the Craig-Gordon model has been extensively discussed).

We don't think it would be helpful to add the exact mathematical expression to our equations because they would be extremely cluttered, we are also not aware this is done in the current literature. Instead, we work under the assumption, that any addition or subtraction involving delta values implies the use of the correct mathematical expression (even for carbon and oxygen data, although expected differences are minor).

In a revised version, we will add this information to the methods section, so that no further confusion shall occur.

Furthermore, we did not use the term "1:1 line" rather we refer to a "1:1 relationship". In our understanding a 1:1 relationship is a simple source-product relationship (i.e. fractionation of a single component according to {Sessions:2005iu}), which is expected during a (simplified) biosynthetic reaction for example, without any other processes affecting it. This doesn't have to lie on a 1:1 line (which in our understanding would have a slope of 1), and for delta values it actually shouldn't, due to the mathematical issues dealing with ratios rather than absolute numbers. Possibly that is the source of confusion.

We will address the other issues raised by Reviewer 1 in a detailed response, when given the opportunity to respond by the editor.

Oliver Rach, Ansgar Kahmen, Achim Brauer and Dirk Sachse.

Rach, O., Brauer, A., Wilkes, H., Sachse, D., 2014. Delayed hydrological response to Greenland cooling at the onset of the Younger Dryas in western Europe. Nature Geoscience 7, 109-112.

Sessions, A.L., Hayes, J.M., 2005. Calculation of hydrogen isotopic fractionations in biogeochemical systems. Geochimica et Cosmochimica Acta 69, 593-597.

─────────────────────────────

---

## Referee Comment (RC3) · Anonymous Referee #3 · 24 Feb 2017

General: Rach et al. use the hydrogen isotopic difference between mid-chain nC23 alkanes and long-chain nC29 alkanes, which they interpret to be mainly derived from aquatic and terrestrial plants, respectively, to infer changes in relative humidity based on a so-called DUB (dual biomarker) model. While I agree that a step forward towards quantitative estimates of changes in terrestrial hydrology based on lipid biomarker hydrogen isotope compositions is needed, I think that the authors underestimate the uncertainties in their approach and underlying assumptions so that the calculated estimates in changes of relative humidity cannot be regarded as precise or even accurate. I agree that the approach should be presented but only with a broader discussion of potential sources of uncertainty.

My main comments are on the assumptions which go into the consideration and the model. Some of them are shortly discussed in the manuscript while others are only 'between the lines'. I think this should be discussed more broadly and openly and would then add to the strength of the paper.

Lipid distributions in plants: The authors assume that the nC23 reflects a signal from the aquatic macrophytes while the nC29 reflects a signal of the integrated terrestrial plant ecosystem. n-Alkane distributions are, however, not so distinctive in plants. Terrestrial plants also make nC23 and macrophytes also make nC29 albeit in smaller amounts. Due to the current lack of isotope data of the smaller abundant compounds it cannot be assumed that the nC23 has the same hydrogen isotope composition as the nC29 in terrestrial plants and macrophytes, respectively. nC29 and nC31 as most abundant alkanes in terrestrial plants often show slightly different hydrogen isotope compositions in the same plant so this would also be expected for nC23 and nC29. In sedimentary mixtures of various alkane sources this is difficult to disentangle. Even if a sediment sample would only contain alkanes from a single plant species such a difference would be interpreted by the model to reflect a difference in evaporative enrichment in leaf waters which would clearly not be the case.

Ecosystem integration: Sediments will collect alkanes from a variety of sources including ones that are derived from distant sources. As alkanes from different plants can have very different hydrogen isotope values depending on used water sources, different biosynthetic fractionation and variable sensitivity to leaf water enrichment any changes in the relative proportions of the supplied alkanes to the sediment, either by changes in the ecosystem composition around the lake or changes in local versus distant sources of alkanes can lead to changes in the recorded signals which have nothing to do with changes in relative humidity at the site. Ecosystem changes might occur due to changing temperature and CO2 levels next to relative humidity. Source water isotopes can change due to shifts in moisture sources and transport pathways. Changes in aeolian-derived alkanes might occur due to changing wind patterns and strengths.

These factors would introduce uncertainty in relative humidity estimates.

Sediment integration: Sediments represent not only spatial but also time-integrated signals. The authors apply their model not to plants but to lipids from sediments which integrate over several years with inter- and intra-annual variability. The investigated sediment samples in Rach et al. (2014) are 1 cm thick. With a sedimentation rate of 0.5 to 3 mm per year in Meerfelder Maar these samples reflect a few to about 20 years at least. The signals recorded by the aquatic and terrestrial lipids could vary from year to year as well as their relative contributions into the sediments which would then lead to signals that are not directly comparable between aquatic and terrigenous lipids regarding the recorded environmental conditions. The signals of both aquatic and terrestrial lipids alone would reflect averaged conditions over the sample integration interval but it seems questionable to me if these are then directly comparable. Although difficult to predict the effect of time-integration might add additional uncertainty to the model results.

Dependence on setting: The authors assume that isotopic enrichment due to lake water evaporation and surface soil water evaporation does not occur. While this may be true for the Meerfelder Maar site it certainly is not true on a larger scale. Surface soil water enrichment occurs in semi-arid to arid areas and shallow-rooting plants incorporate this signal. Lake surface water isotope enrichment occurs in arid areas and then offsets the recorded aquatic signals. Lakes may also be fed by groundwater and can thus be isotopically offset from precipitation. Also the assumption that the isotopic enrichment in terrigenous lipids is due to leaf water enrichment may be questionable on a larger scale. In settings with very short rainy and growing seasons the vegetation might not be sensitive for leaf water enrichment as assumed here. These are clear restrictions of the model to humid regions with rain-fed lakes and should be made clear in the discussion. It can thus not be assumed that the isotopic offset between aquatic and terrestrial solely arises from leaf water enrichment, which in my view is an oversimplification.

In summary, I think the approach to apply a plant physiological model to sedimentary lipid isotope composition is interesting as an exercise to test if the outcome makes sense but highly challenging as sedimentary lipids cannot be treated in a similar fashion as lipids directly derived from plants. The environmental factors regarding variable lipid sources, spatial and temporal integration of signals, and the dependence on the particular setting need to be taken into account and discussed openly. Although the environmental processes which lay between plants and sediments tend to be often ignored in literature it cannot simply be assumed that plant lipids and sedimentary lipids can be treated similarly. An adequate discussion of these environmental processes and associated uncertainties needs to be included. Although likely impossible to quantify, I expect the associated uncertainties to be much larger than the 3.4% in rH based on the model alone probably exceeding the total amplitude of the reconstructed changes in rH. The model results should be discussed in the context of the environmental processes to avoid the risk of an over-interpretation of the model output. In this respect, I wonder if the data derived from the model actually indeed provide more quantitative information than the comparison of the two 'raw' isotopic signals alone as shown in Rach et al. (2014).

---

## Author Comment (AC2) · 18 Apr 2017

We would like to thank the referees for their valuable comments. We are prepared to address and/or clarify all points raised by the reviewers in a revised version of our manuscript. Both reviewer 1 and 3 addressed the problems associated with a general applicability of our approach due to the number of assumptions and parametrizations which we feed into the model. We completely agree with the reviewers on this point and stress that it is not our intention to introduce an approach which is universally applicable. Rather, we see our manuscript as a proof-of-concept, i.e. an idea what is potentially possible, when certain conditions are met and constraints can be made to parametrize the model. We show what we would need to get quantitative information

from biomarker hydrogen isotope data. For this reason, as an example we choose an exceptionally well studied and detailed archive (Lake Meerfelder Maar), where we can constrain the model parameters as accurately as currently possible, and show that we obtain reasonable data, albeit with significant uncertainties. We are prepared to discuss this aspect in a revised version in more detail, so that the manuscript is not mistaken as a blueprint which can be applied to any lacustrine archive. We do understand our contribution as a first step towards quantitative reconstructions from biomarker data.

In the following we provide a point by point answer to the specific comments of referee #1. The original comments of referee #1 you will find below. Our answers are in italic letters below each comment.

Anonymous Referee #1

Rach et al. present a mechanistic leaf-water isotope model to quantify for the first time past changes in relative humidity using lake sedimentary leaf-wax dD records. The model, which is suited for reconstructions from temperate regions where lake water evaporation is negligible, is tested using previously published proxy data from the type site of Meerfelder Maar (MFM) in Western Germany, with a focus on the period spanning the climate transitions into and out of the Younger Dryas stadial. Personally, I'm glad to see this work finally coming along after several years of development. I think this study constitutes a long overdue leap towards a more quantitative estimate of lipid-based stable water isotope reconstructions, which will ultimately be of interest to a broad readership including organic geochemists, paleobotanists, paleoclimatologists and climate modellers.

The construction of the model leans upon a large number of unavoidable assumptions involving both plant physiology and environmental conditions. While the majority of the physiological assumptions are reasonably satisfied through empirical evidence, I think some of the assumptions surrounding past environmental and climate conditions are

less convincing. Nonetheless, due to the lack of sound proxies that allows reconstructing parameters such as local vegetation cover, atmospheric pressure, wind strength and seasonal air temperature (among others), the model formulation presented in this paper should be considered –overall– as good as it can get. However, I think there is still much room for improvement and I have a number of serious concerns, primarily involving the methodological approach, the data-uncertainty treatment, and the contents of the paper that the authors should address before publication.

Main comments 1) I am very surprised the authors decided to design this study directly around downcore data without attempting validation of their model using core-top samples from MFM. I think the model should first be tested using lipid $\delta$D measurements from surface sediments in tandem with meteorological observations before a deglacial Rh reconstruction is discussed. Whether this unusual direction was taken intentionally or not, I think the authors should provide some explanations.

Answer from authors: We do understand the reviewer's concerns, but we deliberately choose this approach. In a general sense, there are two possibilities of validating climate proxy approaches. The most straight forward is to test the proxy under modern conditions and compare results with actual instrumental data. This can be done either along a modern climatological gradient or over the time period where instrumental data are available. Potential problems of this approach are a limited comparability of environmental changes in space and time. The second approach is to use a past time period with known major changes in the parameter to be tested for. We employ the second approach here, because the first approach is unsuitable for the DUB model. This is for the following reasons: 1) Testing the DUB approach with modern core top sediments from MFM (or any other lake) is not feasible, as over the instrumental period (roughly the last 150 years) no major changes in relative humidity occurred. Lake MFM is also not annually laminated (varved) during this period (due to human interference), so that MFM would not be suitable. We don't know of any other lake sediment record which is varved and was subject to major rH changes during the last 150 years. Using only core

top sediments (i.e. only one data point integrating the last decade or so) would not allow for testing the performance of the DUB approach, which aims to reconstruct relative changes in rH, not absolute data (see discussion in the manuscript). 2) We considered testing the DUB approach along a climatic gradient i.e. (Sachse et al., 2004). We concluded this approach to be unsuitable, because we cannot assume that the source of aquatic biomarkers (in our case nC23) is always the same aquatic macrophyte in different lakes and ecosystems. For that matter, it was impossible to find enough lake systems where the sources of aquatic biomarkers are comparable and which cover a large enough aridity gradient. As such, we rejected the modern calibration approach and opted to use a past sediment record with the following constraints: a) Which covers a time period where major changes in relative humidity have been identified before (the YD as the last major abrupt climate change in Europe, see (Brauer et al., 2008; Isarin et al., 1998; Rach et al., 2014) and b) Where sources of in particular aquatic (but also terrestrial) biomarkers can be constrained. In lake MFM detailed pollen records (Brauer et al., 1999) allow this source assignment, as outlined in (Rach et al., 2014), see also the supplementary material of this paper. In brief, during the studied core section of MFM for the aquatic biomarker we observed a covariation of nC23 concentrations and the abundance of Potamogeton remains, so that we concluded nC23 during the study period at MFM is mainly produced by this aquatic macrophyte (Rach et al., 2014). We are aware of the limitations of our approach, in particular it's potential applicability to other records, where such constraints are more uncertain. Our intent is to show that the DUB model is a potential approach to reconstruct quantitative hydrological information, if the sediment record, in particular biomarker source assessments, are well constrainable. In that sense, the well studied MFM YD record is a perfect record to put our hypothesis forward. Our approach is supported by the tight correlation between our $\Delta$rH reconstructions and pollen concentrations of aridity adapted land plants (i.e. Artemisia), see Fig. 4, i.e. two completely independent aridity proxies. In a revised version of the manuscript we would explain this reasoning in more detail and outline the requirements for the applicability in other records. We also outline other potential

approaches, i.e. choosing more source specific aquatic biomarkers (which however are not present in all lake records or not over the whole of the period of interest).

2) One of the main assumptions upon which the results depend is that pollen records are indicative of the local vegetation cover and that pollen counts scale linearly with waxes concentration in MFM sediments. If this were the case, one would expect to observe a clear correlation between the abundance of n-alkanes and pollen records. I would therefore like to see the distribution of n-alkanes plotted together with selected pollen data (e.g. relative distribution of trees, shrubs, herbs). Ideally, the authors should also present some correlation statistics or at the very least a visual correlation using for instance the average n-alkane chain length (and other chain length ratios) to make a qualitative distinction between graminoids and woody plants that would confirm the authors' hypothesis.

Answer from authors: We disagree with the reviewer. The results of our general approach do not at all depend on the assumption that pollen records are indicative of local/regional vegetation. Only for the vegetation correction approaches we rely on this assumption, as we outline also in the paper. We note again, that the vegetation correction does not substantially alters our results, see also below. We do think that at MFM pollen concentrations do record local to regional vegetation, due to the small catchment size of the maar lake etc. (see discussion in (Brauer et al., 1999). However, results from the DUB model with and without (which doesn't rely on the pollen data at all) the vegetation correction shows very similar results. We discuss leaf wax biomarker sources in the manuscript and in more detail in Rach et al. (2014), see supplementary material, where we also show pollen and leaf wax concentrations. As such, we don't think it is necessary to provide this here again. Correlations between the amount of biomarkers and the amount of pollen would not support the assumption, as different plant types produce different amounts of n-alkanes. As such, an increase in grass pollen may be accompanied by a decrease or an increase in n-alkane concentrations, if grasses produce less or more n-alkanes respectively, than the previously dominant

species.

3) The authors argue that monocots integrate a varying evaporative $\delta$D enrichment signal in response to changes in humidity conditions. Following this reasoning, the mixing ratio between leaf water and un-enriched xylem water in C3 grasses will change as a function of the aridity level in the catchment. Therefore, the weighing of the fraction of grass cover used to correct $\varepsilon$terr-aq should also vary over time, whereby a much higher fraction of C3 grass integrated the leaf-water evaporative $\delta$D enrichment during the Late Allerød (AL) and Early Holocene, when climatic conditions were presumably wetter. The authors instead apply a constant weighing factor of 18% throughout the record in their **correction. This issue should be addressed and the weighing factor should vary over time according to the humidity conditions as inferred using an independent proxy (for instance Artemisia pollen percentages).

Answer for authors: The comment above is based on a misunderstanding and we will clarify this in a revised version of our manuscript. We did not apply a constant weighing factor in our **correction (but in our *correction we did). The model weighs the leaf water isotope enrichment (i.e. $\varepsilon$terr-aq) with only 18% for the fraction derived from grasses (estimated through the pollen record). This results in a lower reconstructed relative humidity during the dry YD, as opposed to the * and uncorrected model versions, since according to the hypothesis which forms the base of this correction, leaf water enrichment under dryer conditions is larger than recorded in grass leaf wax n-alkane $\delta$2H values, as opposed to wetter conditions (see discussion in Kahmen et al. 2013 and in line 499-522 in our manuscript). That no constant weighing factor was applied in the **correction (as opposed to the *correction) is also visible in Fig. 3, where the * and the **correction show different degrees of changes, in particular during the YD. If ** would have used a constant weighing (as was used for *), then both curves should covary and feature a constant offset.

4) Despite the authors' effort to address all the potential errors that accompany their data, there is a large source of uncertainty that has been neglected and that has been

allegedly circumvented by simply including the analytical uncertainty of the $\delta$D measurements. This source of uncertainty is the variability associated with the apparent isotope fractionation values of C3 dicots and C3 monocots as observed for modern plants (e.g. Sachse et al., 2012). This variability should generally be accounted for when estimating Rh. It should also be taken into consideration when calculating the vegetation corrected $\varepsilon$terr-aq*, as the authors simply apply the mean $\varepsilon$app difference between dicots and monocots disregarding the related uncertainty, which can be quite substantial (as high as 30‰ at 1 sigma) for each plant group. While I understand that quantifying this uncertainty would disproportionally increase the error bounds of their Rh reconstruction, I think the authors should at least discuss this issue more openly in the text. Furthermore, along these lines I believe that the authors should attempt at estimating how well the pollen records represents the local vegetation cover and include this source of uncertainty in the error propagation equation.

Answer from authors: Firstly, we note that this discussion only affects the reconstructions derived from the "vegetation corrected" DUB* model (not the DUB** or the not vegetation corrected results). We respectfully, but decisively disagree with the reviewer, that adding the variability of the apparent fractionation (i.e. the isotopic difference between source water and n-alkanes) would result in more accurate estimations of DUB* model uncertainty. This is because the observed variability in the apparent fractionation for higher plants is due to at least two different processes: a) differences in the amount of plant transpiration, i.e. leaf water isotope enrichment during leaf wax synthesis, which likely cause the larges variability b) differences in biosynthetic fractionation among different species see (Kahmen et al., 2013a; Kahmen et al., 2013b; Sachse et al., 2012)

With the DUB approach we actually account for the large part of the variability caused by these processes: we assume that $\delta 2$Hterr does not represent source water, but leaf water of plants, as such removing the variability caused by process a) from the model. As of now, we have not enough information to estimate the amount of variability origi-

nating from process b), but it's likely significantly smaller than from a) in particular when only a limited number of species are n-alkane producers. Because of this uncertainty we also used another approach for obtaining "vegetation corrected" results (DUB**), which relies on a completely different hypothesis (see above). We note that all three DUB model runs (with and without vegetation correction) show quite similar results. So we rather see the range of results from the 3 approaches as the current uncertainty of model results (we state this in line 563): "Tentatively, the lower variability in $\Delta$rh** within the YD as well as the less pronounced shift in particular at the onset and termination of the YD (Fig. 3A) provides are more realistic scenario. But as of now, we regard the differences in predictions as the error of quantitative predictions from the DUB approach"

5) I am quite sceptical about the use of chironomid-based temperature reconstructions to infer air temperatures. Although I recognize that as yet lacustrine midge assemblages are one of the best proxies for summer air temperature, they have shown to incorporate a number of environmental signals apart from air temperature alone, most importantly catchment vegetation, nutrient levels, lake depth and seasonality (Eggermont and Heiri, 2012; Luoto, 2010). It is therefore likely that this biological proxy was very sensitive to the major environmental and seasonality shifts that occurred at the onset and termination of the YD. Especially, colder and longer winters during the YD might have resulted in relatively colder surface water temperatures during the growing season relative to the air (e.g. replenishment of the local aquifer via snow thawing), in contrast to the preceding warm AL phase. As chironomids are sensitive to water temperatures, it is hard to say to what extent the proxy is biased towards colder temperatures during the YD, not mentioning that the authors use a Dutch record thus making impossible to assess local versus regional factors. I think some of these issues should be openly discussed in the paper. Moreover, if I remember correctly the chronology that underpins the record from Hijkermeer is based on 14C dating of regional pollen boundaries that have been dated elsewhere. This implies that the temperature record comes with fairly large age uncertainties as compared to the proxy records at MFM,

where the chronology is much more accurate. Assuming that the alignment between Hijkermeer pollen records and the regional pollen stratigraphy is precise, I wonder if the author can include in their error propagation estimate the age uncertainty associated with the temperature reconstruction. Even though I understand temperature plays only a minor role in the DUB model, in my opinion, this would result in a much more rigorous estimation of the true uncertainties that accompany the reconstructed Rh. In addition, I suggest that for reference the authors plot the temperature record together with the Rh reconstruction.

Answer from authors: We are aware of the possible uncertainties of chironomid-based temperature reconstruction to infer air temperatures, as well as of the age model uncertainties and local vs. regional differences, since we compare 2 different sites. The uncertainties in temperature estimations, including a potential seasonal bias, are somewhat included in the uncertainty of the results, which we propagated into the DUB model. Of course, ideally a local T record should be used, but as none is available from MFM, we used the closest record. Also here we are interested in relative changes, i.e. differences between temperatures before and during the YD. Newer studies suggest that these temperature patterns are spatially different between N, Central and S Europe, but very consistent within these regions, which is also supported by modelling exercises (Heiri et al., 2014). As such, we see it as a well supported assumption, that temperature differences between the YD and the preceding and following time periods are well represented by this Hijkermeer record.

The age model uncertainty (from the Hijkermeer 14C based age model) is implicitly included in our calculations, due to the different temporal resolution of both records (Hijkermeer record consists of 37 datapoints, whereas the our MFM $\varepsilon$terr-aq dataset of 106 datapoints). Therefore we calculated an equidistant time series for comparison, which results in a (time) averaged record of the lower resolution dataset. We don't see this loss of resolution as problematic, as also here we are interested in relative changes, i.e. differences before, after and during the YD. We can add this more

explicitly to the method section, but refrain from further assessing the potential uncertainties in chironomid based temperature reconstructions, as this methodology has been covered elsewhere and validated in particular for temporal differences during the Late Glacial period with climate model data (Heiri et al., 2014).

6) Since the authors decided to reconstruct Rh across the Younger Dryas, I think it would be appropriate to briefly present the status of the knowledge on this climate event (as well as to better frame their results into a paleoclimatological context). I suggest to mention the ocean–sea-ice–atmosphere mechanisms that would explain the climate variability observed in European climate reconstructions during this period (e.g. Brauer et al., 2008; Lane et al., 2013; Muschitiello et al., 2016; Rach et al., 2014). I would also recommend that the authors discuss the current understanding of hydroclimatological variability at the onset and termination of the YD in Europe based on the lake sedimentary $\delta$D and $\Delta\delta$Dterr-aq reconstructions available so far.

Answer from authors: We believe an extensive discussion or analysis of the current understanding of hydroclimate variability is not in the scope of our manuscript, as we don't conclude on the mechanistic drivers of these processes here, which is done elsewhere in the literature, but rather present a new approach to estimate quantitative hydrological data. Therefore we refrain from further adding a discussion in our manuscript.

Specific comments The DUB is based on a number of important assumptions that are discussed along the text (I counted at least 14 in the paper, some of which are "hidden" between the lines). I wonder if the author can provide a summary of these assumptions in the form of a table to facilitate rapid screening. Similarly, I suggest that all the model parameters, fixed variables, and sensitivity tests are summarised in separate tables. As they are, these information are hard to piece together.

Answer from authors: We agree with the reviewer, and will provide a table with important assumption and fixed parameters in the supplement of a revised version of our manuscript.

L170: The authors adopt a constant atmospheric pressure value in their model. I would first like to know what value they use and why? Secondly, I would like to know how sensitive their model is to this parameter. Climate modelling studies have shown that there were considerable summer sea level pressure changes over Northern-Central Europe from the late AL to the YD (Menviel et al., 2011; Muschitiello et al., 2015). I would therefore be inclined to apply different sea-level pressure values across the deglaciation. Perhaps the authors can comment on this and openly discuss these problems in the paper.

Answer from authors: We applied a constant value for atmospheric pressure of 971 hPa, which is based on the barometric formula, i.e the mean atmospheric pressure at the elevation of MFM at 337m above sea level (pressure at sea level 1013 hPa), see line 207-208. The elevation of the lake has not changed during the investigated period and the model sensitivity to this value is very low (i.e. 0.05% rH change for a 100hPa changes, which encompasses the highest and lowest pressure ever measured in Germany for example), so that short-term weather related changes would not impact the results.

L176-182: Is there any empirical value that allows calculating Tleaf as a function of Tair? For the reasons I outlined in the previous comments, assuming that it is constantly (and equally) cloudy and/or windy at MFM during both AL and YD does not necessarily hold.

Answer from authors: Unfortunately there are no long-term data on this, but studies show that air and leaf temperature during daily cycles have a nearly 1:1 correlation for a temperature range between 15 – 20°C (Kahmen et al., 2011), but can deviate more at higher temperatures. Since during the cold YD the environmental temperature would not have exceeded this range significantly and our reconstruction datapoints integrate over a decade, we think this assumption is justified.

L487-488: A number of studies have shown a bi-partite structure of the YD with relatively drier conditions in Northern, Central and Southern Europe during the Early YD and relatively drier conditions during the Late YD (Bakke et al., 2009; Bartolomé et al., 2015; Lane et al., 2013). It surprises me that this mid-YD transition is not clearly captured in the Rh reconstruction at MFM. Although the authors claim that the record reveals "centennial scale excursions to higher $\Delta$Rh after 12.100 BP" I struggle to see any appreciable change in Rh variability. Critically, a marked shift in Rh after the mid-YD transition would support the reconstructed Rh, since virtually no significant vegetation shift had occurred during the YD and therefore the modelled Rh is independent of potential influences from local vegetation changes during this period. However, I must acknowledge that so far the mid-YD transition has been inferred only using qualitative or indirect hydro-climate proxies and thus a net shift from dry to wet conditions in Europe still requires conclusive evidence. Perhaps these issues can be briefly addressed in an apposite YD section of the paper (please see main comment on YD background discussion).

Answer from authors: We are aware that $\varepsilon$terr-aq and also $\Delta$rH reconstruction do show different mean values before and after 12.100 BP, i.e. we do reconstruct a slightly higher rH after 12.100 BP for all model runs (on average 3% higher). This 2-phased YD is also reflected in a variety of other proxies at MFM, for example the decrease of Artemisia pollen after 12.100 BP. Since the change in $\Delta$rH is within our model uncertainty, we refrained from including this observation.

L491-494: How does the percentage of shrub pollen vary with respect to the percentage of tree and herb. If there is a strong covariance between shrubs, trees and herbs then it is not surprising that both the vegetation-corrected (using grass and tree+grass, respectively) Rh reconstructions correlate with the Artemisia pollen percentage (i.e. included in the shrubs pollen record) better than the uncorrected Rh. I would therefore like to know the level of covariance between the distributions of trees, grasses and shrubs at MFM. In addition, I would recommend that the authors include the relative shrub pollen percentages in Figure 3 for reference (please note that in the same figure either tree or herb distributions have not been plotted). I also wonder whether it is possible that the improved fit between the corrected Rh and Artemisia data (Figure 4) merely stems from subduing the Rh series variability when applying the vegetation correction. I believe that the paper would benefit from including some selected pollen diagrams (as supplementary material for example). Analogously, I think the original $\delta$D and $\Delta\delta$Dterr-aq records should also be presented in Figure 2 or 3. I also recommend that the author consider to include as Figure 1 their conceptual overview model of the hydrogen-isotopic relationship between source water and sedimentary lipids (Figure 6 in Sachse et al. (2012) and Figure S6 in Rach et al. (2014)) to illustrate the initial formulation steps of the DUB model.

Answer from authors: As mentioned in our manuscript (line 462-466) we assume Betula and Salix as the most dominant tree vegetation during the YD in the catchment of MFM from available pollen data. Only these two species were included in the calculation of the tree fraction (ftree) (see line 464). We did not use any "shrub" classification for any of our calculations (there is also no such classification existing for MFM in the literature), but only a "grass" classification for quantifying the fraction of grasses (fgrass). Since Artemisia does also not belong to grass classification (i.e. is not included in fgrass) the Artemisia record represents an independent proxy which therefore has no direct or indirect influence on the vegetation corrections.

Furthermore, we don't see any necessities to include to the original $\delta$2H data in our manuscript since these data are already published and extensively discussed previously (Rach et al., 2014). We would include a plot of $\varepsilon$terr-aq as our main model input parameter. We also think that the manuscript would not benefit from a plot of the conceptual overview as suggested by the referee since these plots are already published several times (as mentioned by the reviewer). To better illustrate the model input parameters, we will include a conceptual overview on our modelling approach (similar to Figure 1 in Kahmen et al. 2011) in a revised version of our paper.

Line-by-line comments

Answer from authors: Issues below will be fixed in a revised version of the manuscruipt.

L77-78 and 87-88: In equations (1) and (2) please specify that the terms $\varepsilon$bio refer to terrestrial and aquatic components, respectively (i.e. $\varepsilon$bio (terr) versus $\varepsilon$bio (aq)).

L159: The term esat "Saturation vapour pressure" should be introduced at line 148.

L172: Missing "of" after "function".

L243-244: Please provide reference for this statement.

L392: The line in brackets should start with small letters.

L416: "... low humidity treatment": how much?

L462: "...provides are more ... " should read " ... provides a more ... "

Figure 1: The data plotted in the upper-right panel are not in scale with the data presented in the upper-left panel. Please adjust.

Figure 3: Either the tree or shrub relative distribution is missing from the figure.

Brauer, A., Endres, C., Günter, C., Litt, T., Stebich, M., Negendank, J.F.W., 1999. High resolution sediment and vegetation responses to Younger Dryas climate change in varved lake sediments from Meerfelder Maar, Germany. Quaternary Science Reviews 18, 321-329.

Brauer, A., Haug, G.H., Dulski, P., Sigman, D.M., Negendank, J.F.W., 2008. An abrupt wind shift in western Europe at the onset of the Younger Dryas cold period. Nature Geoscience 1, 520-523.

Heiri, O., Brooks, S.J., Renssen, H., Bedford, A., Hazekamp, M., Ilyashuk, B., Jeffers, E.S., Lang, B., Kirilova, E., Kuiper, S., Millet, L., Samartin, S., Toth, M., Verbruggen, F., Watson, J.E., van Asch, N., Lammertsma, E., Amon, L., Birks, H.H., Birks, H.J.B., Mortensen, M.F., Hoek, W.Z., Magyari, E., Muñoz Sobrino, C., Seppä, H., Tinner, W., Tonkov, S., Veski, S., Lotter, A.F., 2014. Validation of climate model-inferred regional

temperature change for late-glacial Europe. Nat Commun 5.

Isarin, R.F.B., Renssen, H., Vandenberghe, J., 1998. The impact of the North Atlantic Ocean on the Younger Dryas climate in northwestern and central Europe. Journal of Quaternary Science 13, 447-453.

Kahmen, A., Hoffmann, B., Schefuss, E., Arndt, S.K., Cernusak, L.A., West, J.B., Sachse, D., 2013a. Leaf water deuterium enrichment shapes leaf wax n-alkane delta D values of angiosperm plants II: Observational evidence and global implications. Geochimica et Cosmochimica Acta 111, 50-63.

Kahmen, A., Sachse, D., Arndt, S.K., Tu, K.P., Farrington, H., Vitousek, P.M., Dawson, T.E., 2011. Cellulose delta(18)O is an index of leaf-to-air vapor pressure difference (VPD) in tropical plants. Proceedings of the National Academy of Sciences 108, 1981-1986.

Kahmen, A., Schefuss, E., Sachse, D., 2013b. Leaf water deuterium enrichment shapes leaf wax n-alkane delta D values of angiosperm plants I: Experimental evidence and mechanistic insights. Geochimica et Cosmochimica Acta 111, 39-49.

Rach, O., Brauer, A., Wilkes, H., Sachse, D., 2014. Delayed hydrological response to Greenland cooling at the onset of the Younger Dryas in western Europe. Nature Geoscience 7, 109-112.

Sachse, D., Billault, I., Bowen, G.J., Chikaraishi, Y., Dawson, T.E., Feakins, S.J., Freeman, K.H., Magill, C.R., McInerney, F.A., van der Meer, M.T.J., Polissar, P., Robins, R.J., Sachs, J.P., Schmidt, H.-L., Sessions, A.L., White, J.W.C., West, J.B., Kahmen, A., 2012. Molecular Paleohydrology: Interpreting the Hydrogen-Isotopic Composition of Lipid Biomarkers from Photosynthesizing Organisms. Annual Review of Earth and Planetary Sciences 40, 221-249.

Sachse, D., Radke, J., Gleixner, G., 2004. Hydrogen isotope ratios of recent lacustrine sedimentary n-alkanes record modern climate variability. Geochimica et Cosmochimica Acta 68, 4877-4889.

---

## Author Comment (AC3) · 18 Apr 2017

We would like to thank the referees for their valuable comments. In the following we provide a point by point answer to the specific comments of referee #3. The original comments of referee #3 you will find below. Our answers are in italic letters below each comment.

**Anonymous Referee #3**

General: Rach et al. use the hydrogen isotopic difference between mid-chain nC23 alkanes and long-chain nC29 alkanes, which they interpret to be mainly derived from aquatic and terrestrial plants, respectively, to infer changes in relative humidity based

on a so-called DUB (dual biomarker) model. While I agree that a step forward towards quantitative estimates of changes in terrestrial hydrology based on lipid biomarker hydrogen isotope compositions is needed, I think that the authors underestimate the uncertainties in their approach and underlying assumptions so that the calculated estimates in changes of relative humidity cannot be regarded as precise or even accurate. I agree that the approach should be presented but only with a broader discussion of potential sources of uncertainty. My main comments are on the assumptions which go into the consideration and the model. Some of them are shortly discussed in the manuscript while others are only 'between the lines'. I think this should be discussed more broadly and openly and would then add to the strength of the paper.

Lipid distributions in plants: The authors assume that the nC23 reflects a signal from the aquatic macrophytes while the nC29 reflects a signal of the integrated terrestrial plant ecosystem. n-Alkane distributions are, however, not so distinctive in plants. Terrestrial plants also make nC23 and macrophytes also make nC29 albeit in smaller amounts. Due to the current lack of isotope data of the smaller abundant compounds it cannot be assumed that the nC23 has the same hydrogen isotope composition as the nC29 in terrestrial plants and macrophytes, respectively. nC29 and nC31 as most abundant alkanes in terrestrial plants often show slightly different hydrogen isotope compositions in the same plant so this would also be expected for nC23 and nC29. In sedimentary mixtures of various alkane sources this is difficult to disentangle. Even if a sediment sample would only contain alkanes from a single plant species such a difference would be interpreted by the model to reflect a difference in evaporative enrichment in leaf waters which would clearly not be the case.

Answer from authors: We are well aware of the issues the reviewer touches here. In that sense, MFM as a well studied site does allow us to constrain lipid sources to a degree which is likely more difficult to obtain at other sites. We are prepared to outline our general aims with the model, the problems with a universal applicability (see answer to Rev. #1) as well as the assumptions and model parametrizations in more detail in a revised version (i.e. a table with the assumptions etc., see answers to Rev. #1). With regard to source assignments of biomarkers: we did discuss lipid sources and how they can be constrained at MFM in detail in the supplementary material of (Rach et al., 2014), and believe that we should not repeat the whole discussion here. In brief, the detailed pollen record allows for biomarker source assessment. There we also show the tight covariation between nC29 and nC31 alkane  $\delta$ 2H values, which we regard as being derived from a similar terrestrial source. These two compounds have similar absolute  $\delta$ 2H values, whereas nC23  $\delta$ 2H values are more enriched, suggesting a different source (i.e. macrophyte).

Ecosystem integration: Sediments will collect alkanes from a variety of sources including ones that are derived from distant sources. As alkanes from different plants can have very different hydrogen isotope values depending on used water sources, different biosynthetic fractionation and variable sensitivity to leaf water enrichment any changes in the relative proportions of the supplied alkanes to the sediment, either by changes in the ecosystem composition around the lake or changes in local versus distant sources of alkanes can lead to changes in the recorded signals which have nothing to do with changes in relative humidity at the site. Ecosystem changes might occur due to changing temperature and CO2 levels next to relative humidity. Source water isotopes can change due to shifts in moisture sources and transport pathways. Changes in aeolian-derived alkanes might occur due to changing wind patterns and strengths. These factors would introduce uncertainty in relative humidity estimates.

Answer from authors: We agree with the reviewer on the potential relevance of these processes, but we can either rule them out or we account for them: We do discuss ecosystem changes (which are reconstructed from pollen data) as potential factors influencing our model outcome (see also the supplementary material of Rach et al. 2014) and in the present manuscript we present an approach to quantify the influence on the model outcome (vegetation correction). During the YD no major changes in atmospheric CO2 occurred, as such we can rule this process out. Source water  $\delta$ 2H values

СЗ

can (and likely did) change due to moisture source changes during the onset and termination of the YD, but our approach, i.e. using terrestrial and aquatic biomarkers, which use the same source water, is therefore not dependent on this factor. In addition, the lake's small catchment area (5,76 km2) with steep, vegetated crater walls, sheltering the lake from wind makes a dominant input from more distant sources unlikely vs. the importance of local sources.

Sediment integration: Sediments represent not only spatial but also time-integrated signals. The authors apply their model not to plants but to lipids from sediments which integrate over several years with inter- and intra-annual variability. The investigated sediment samples in Rach et al. (2014) are 1 cm thick. With a sedimentation rate of 0.5 to 3 mm per year in Meerfelder Maar these samples reflect a few to about 20 years at least. The signals recorded by the aquatic and terrestrial lipids could vary from year to year as well as their relative contributions into the sediments which would then lead to signals that are not directly comparable between aquatic and terrigenous lipids regarding the recorded environmental conditions. The signals of both aquatic and terrestrial lipids alone would reflect averaged conditions over the sample integration interval but it seems questionable to me if these are then directly comparable. Although difficult to predict the effect of time-integration might add additional uncertainty to the model results.

Answer from authors: Again, we agree with the reviewer that these processes can be important. But at MFM, we can rule out several of these effects due to the small catchment size and steep morphology: the residence time of n-alkanes in this small and steep catchment (i.e. from leaf or soil) is likely short, we assume within our sampling resolution (i.e. decades). In this catchment we assume that potential differences in the temporal integration between aquatic and terrestrial biomarkers are small, i.e. smaller than our sampling resolution (decadal on average), which is supported by the similar sample to sample (i.e. decadal) variability in their lipid  $\delta$ 2H values. If, for example, terrestrial leaf wax n-alkanes would have a substantially longer residence time in the

soils before being transported into the lake, then the decadal variability should be much smaller, as the soil would already deliver a more integrated signal into the lake. We will outline this argument in a revised version of the manuscript.

Dependence on setting: The authors assume that isotopic enrichment due to lake water evaporation and surface soil water evaporation does not occur. While this may be true for the Meerfelder Maar site it certainly is not true on a larger scale. Surface soil water enrichment occurs in semi-arid to arid areas and shallow-rooting plants incorporate this signal. Lake surface water isotope enrichment occurs in arid areas and then offsets the recorded aquatic signals. Lakes may also be fed by groundwater and can thus be isotopically offset from precipitation. Also the assumption that the isotopic enrichment in terrigenous lipids is due to leaf water enrichment may be questionable on a larger scale. In settings with very short rainy and growing seasons the vegetation might not be sensitive for leaf water enrichment as assumed here. These are clear restrictions of the model to humid regions with rain-fed lakes and should be made clear in the discussion. It can thus not be assumed that the isotopic offset between aquatic and terrestrial solely arises from leaf water enrichment, which in my view is an oversimplification.

Answer from authors: We agree with the Reviewer on the relevance of these issues, as these were part of the reason why we applied to model to the MFM record. In a revised version of the manuscript we will expand the discussion on the problems of a universal applicability of the model under different environmental and hydroclimatic boundary conditons. See also answers to Rev. #1.

In summary, I think the approach to apply a plant physiological model to sedimentary lipid isotope composition is interesting as an exercise to test if the outcome makes sense but highly challenging as sedimentary lipids cannot be treated in a similar fashion as lipids directly derived from plants. The environmental factors regarding variable lipid sources, spatial and temporal integration of signals, and the dependence on the particular setting need to be taken into account and discussed openly. Although the

environmental processes which lay between plants and sediments tend to be often ignored in literature it cannot simply be assumed that plant lipids and sedimentary lipids can be treated similarly. An adequate discussion of these environmental processes and associated uncertainties needs to be included. Although likely impossible to quantify, I expect the associated uncertainties to be much larger than the 3.4% in rH based on the model alone probably exceeding the total amplitude of the reconstructed changes in rH. The model results should be discussed in the context of the environmental processes to avoid the risk of an over-interpretation of the model output. In this respect, I wonder if the data derived from the model actually indeed provide more quantitative information than the comparison of the two 'raw' isotopic signals alone as shown in Rach et al. (2014)

Answer from authors: As mentioned in the 2nd comment we are aware of these limitations and choose the MFM record for exactly those reasons, as we can rule out a number of these processes. We again state, that we do not believe the DUB model is universally applicable, but only at locations, where these boundary conditions are met (we plan to provide a checklist of boundary conditions for applicability in the revised version). The calculated uncertainty of 3.4% includes all errors which can be quantified in some way, but likely not the full uncertainty of the model due to the incomplete understanding of some processes. That is why we rather see the range of results from the 3 approaches (uncorrected and corrected) as the current uncertainty of model results (we state this in line 563): "Tentatively, the lower variability in  $\Delta rh^{**}$  within the YD as well as the less pronounced shift in particular at the onset and termination of the YD (Fig. 3A) provides are more realistic scenario. But as of now, we regard the differences in predictions as the error of quantitative predictions from the DUB approach".

Rach, O., Brauer, A., Wilkes, H., Sachse, D., 2014. Delayed hydrological response to Greenland cooling at the onset of the Younger Dryas in western Europe. Nature Geoscience 7, 109-112.

---

## Author Response (AR1)

Dear Dr. McClymont

We would like to thank you very much for your constructive comments on our manuscript. We improved our manuscript by following your and the reviewers suggestions. Please find a point by point reply on your comments below (*italic letters*).
As already commented earlier in our point by point reply to each reviewer we also took into account the critical comments by the reviewers.

The manuscript by Rach et al. presents a new approach to assessing palaeo-hydrology changes using the co-registered stable hydrogen isotope signatures of "aquatic" and "terrestrial" n-alkanes. The authors formulate a model whereby changes in relative humidity can be isolated from the sedimentary record, using a combination of the stable isotope data and a plant modelling approach. Overall the reviewers are positive about the new insights which can be gained from the approach proposed here. A number of concerns have been raised across the three reviews, however, which indicate that revisions are required to ensure that the practical steps and assumptions made by the authors are clarified, and that the interpretations are robust as a result.

In most instances, the authors have indicated in their reply that they can address the concerns of the reviewers. Of key importance is that the uncertainties of the approach are discussed in more detail (perhaps splitting the current section 3 into 3.1 Uncertainties and 3.2 Sensitivity tests). I agree with the reviewers that there are uncertainties that are also difficult to quantify (e.g. transport of the lipids), but which likely increase the errors on the reconstructions. But these should still be clearly stated. The authors note that many uncertainties have been accounted for or minimised given the particular location of the lake being studied here: it is also critical that such information is more clearly outlined.

***Answer from authors:***
*As suggested by the editor we split Section 3 in '3.1 – Uncertainties' and '3.2 – Sensitivity tests'. In Section 3.1 we now clearly state that there is a difference between 'quantifiable errors' which can be quantified by an error propagation function and 'non-quantifiable uncertainties' which can increase the error of the model output and need to be taken in consideration before applying the model to a certain catchment or record. As you state above, these can be minimized through site selection (i.e. a small catchment lake, from a temperate climate with excellent high-resolution paleoclimate data). We now state this more clearly (lines 259-267). In the revised Section 2 'Approach and Model' (line 124-143) we discuss these non-quantifiable assumption regarding lake and catchment conditions. We clearly state in this part that the application of the DUB approach requires a good understanding of the paleolake system and it's environment. Additionally we added a table (Table 1) in the Appendix where we provide an overview on the uncertainties and assumptions of the DUB approach with specific explanations. Furthermore, we added another Figure (new Fig. 1) which shows a schematic overview on the DUB approach model and the functional relationships between model variables. We also extended the 'conclusion' section (line 601-610) by adding a statement that the DUB approach provides only a first step towards improvement of quantitative analysis of hydrological variability and is not universally applicable.*

In addition to the comments provided by the Reviewers, I am keen to see a note in the section on sensitivity tests (currently lines 239-244) giving some indication of whether these tested variations in each value are considered realistic/feasible, or are they at the upper/lower end of feasibility? (are they specific to this site?) This would help the reader to understand whether the sensitivity tests are giving an over- or under-estimation of the errors.

***Answer from authors:***
*In the section 'Sensitivity tests' we now provide information and reference on the selected range of the sensitivity test for the different climatic regions (line 290-298).*

Drawing on the information given by Reviewer 3, it is also important that the universal applicability of this study (or not) is more clearly highlighted: does MFM provide such a special case that this is a unique record? If such an approach was to be applied elsewhere, what information would be required?

***Answer from authors:***
*As suggested by the reviewers we revised and expanded the section 4 'Application: Reconstructing quantitative changes in Δrh during the Younger Dryas (YD) in Western Europe'. In the first part of this section (line 321-337) we explain our rationale on why we chose a paleo record for model validation instead of modern core top sediments. We also explain why the sedimentary stable isotope records from Meerfelder Maar in combination with high-resolution vegetation data and the regional catchment conditions provide ideal conditions to apply the model on this record (line 345-351).*

I disagree with the authors replies in a couple of instances, and would like them to consider these comments in their revised manuscript:

1) Display of the original n-alkane d2H data in figure 2. This is the first step towards generating the rh output, and I think it is important that the reader can see how this data developed. As the manuscript currently stands, it has to be read with Rach et al. (2014) open at the same time. It does no harm (and makes life easier) to show some the key data in its original form. Likewise, I encourage the authors to show the chironomid data on either figure 2 or figure 3, especially if it is this data which supports their inference that temperature did not fall below 15*C (see page C10 of reply to reviewer 1). The chironomid data is part of the input to the model, and given the discussions about the resolution and age uncertainties it is important for this to be directly compared against the rh model output.

*Answer from authors:*

*As suggested by the editor we revised Figure 2 (now Fig. 3) by adding the original $\delta^2H$ values of the aquatic and terrestrial plant derived n-alkanes from Meerfelder Maar, the terrestrial evapotranspiration record ($\varepsilon_{terr-aq}$) as major input variable of the DUB approach and the original and interpolated temperature record from Hijkermeer. The dataset from Kahmen et al 2011b shows that for temperatures between 15-20℃ air temperature equals leaf temperature. There are no datasets for lower temperatures, but the potential divergence between $T_{leaf}$ and $T_{air}$ is expected to be lower at lower temperatures. However we want to point out that even for a high-resolution multiproxy paleo-dataset (i.e. YD at MFM) the temporal integration is at minimum 9 years. Any slight differences between air temperature and leaf temperature on a daily, weekly or monthly mean value are not relevant due to the high temporal integration.*

2) Reviewer 1 asked for some information about the palaeo-hydrology of the Younger Dryas event here (comment 6). I disagree that nothing can be said about this here: a short paragraph which notes the main patterns of hydrological change would provide a valuable rationale for why this event was targeted. MFM is a key site for this time interval, and this could be emphasised. Since the new results seem to support the existing interpretations of changing humidity, a detailed discussion and interpretation of the climate drivers is not required.

*Answer from authors:*

*We added a short discussion/ summary on the climatological, hydrological and ecological changes with specific references to section 4 'Application: Reconstructing quantitative changes in Δrh during the Younger Dryas (YD) in Western Europe' (line 337-345).*

On page 2 there is a discussion about how the stable isotope approach has developed and been applied to understand palaeo-hydrology. I missed a note to the peatland work of Nichols, who has also combined aquatic and higher plant deuterium/hydrogen ratios as a way of understanding bog hydrology (e.g. Nichols et al. 2009 Quantitative assessment of precipitation seasonality and summer surface wetness using ombrotrophic sediments from an Arctic Norwegian peatland. Quaternary Research)

*Answer from authors:*

*As suggest by the editor we added a short revised discussion on the current state of quantitative analysis and reference to the work of Nichols et al. 2009 (line 54-60)*

Minor typos or corrections:
Line 57: and thus allow the extraction of quantitative..
Line 67: terrestrial plants
Line 70: organisms
Lines 162-164, lines 173-175: where are the values provided in the equations sourced from? No references are given.
Lines 200-201: we did not, however, …
Line 291: since no paleotemperature…
Line 412: italicise n- of n-alkanes
Line 420: hypothesis would imply that leaf
Line 426: move 'also' to after 'would' (before 'underestimate')
Line 451: 'place' rather than 'level'?
Line 459: but a lack of mechanistic understanding
Figure 3: is the Younger Dryas the area shaded in grey?

Line 483: drier conditions
Line 484: remove comma and formulate citation into one set of brackets

*Answer from authors:*
*Suggested typos are corrected and references have been added to the manuscript.*

**Additional comments of referees #1, #2 and #3 which were also taken into account:**

- *the model output has been tested for a variation of the atmospheric pressure ($e_{atm}$) of +/- 100hPa. The results only show an effect of 0,05% on the calculated change of relative humidity. (A statement on that has been added to the section 'Sensitivity tests' (line 305-306)) This effect is significantly below the error range.*

[revised manuscript text omitted]